# Systematic Analysis of Cluster Similarity Indices: How to Validate Validation Measures

## Abstract

There are many cluster similarity indices used to evaluate clustering algorithms, and choosing the best one for a particular task remains an open problem. We demonstrate that this problem is crucial: there are many disagreements among the indices, these disagreements do affect which algorithms are chosen in applications, and this can lead to degraded performance in real-world systems. We propose a theoretical solution to this problem: we develop a list of desirable properties and theoretically verify which indices satisfy them. This allows for making an informed choice: given a particular application, one can first select properties that are desirable for a given application and then identify indices satisfying these. We observe that many popular indices have significant drawbacks. Instead, we advocate using other ones that are not so widely adopted but have beneficial properties.

## 1 Introduction

*Clustering* is an unsupervised machine learning problem, where the task is to group objects that are similar to each other. In network analysis, a related problem is called *community detection*, where grouping is based on relations between items (links), and the obtained clusters are expected to be densely interconnected. Clustering is used across various applications, including text mining, online advertisement, anomaly detection, and many others (Allahyari et al., 2017; Xu & Tian, 2015).

To measure the quality of a clustering algorithm, one can use either internal or external measures. *Internal measures* evaluate the consistency of the clustering result with the data being clustered, e.g., Silhouette, Hubert-Gamma, Dunn, and many other indices. Unfortunately, it is unclear whether optimizing any of these measures would translate into improved quality in practical applications. *External* (cluster similarity) *measures* compare the candidate partition with a reference one (obtained, e.g., by human assessors). A comparison with such a gold standard partition, when it is available, is more reliable. There are many tasks where external evaluation is applicable: text clustering (Amigó et al., 2009), topic modeling (Virtanen & Girolami, 2019), Web categorization (Wibowo & Williams, 2002), face clustering (Wang et al., 2019), news aggregation (see Section 3), and others. Often, when there is no reference partition available, it is possible to let a group of experts annotate a subset of items and compare the algorithms on this subset.

Dozens of cluster similarity measures exist and which one should be used is a subject of debate (Lei et al., 2017). In this paper, we systematically analyze the problem of choosing the best cluster similarity index. We start with a series of experiments demonstrating the importance of the problem (Section 3). First, we construct simple examples showing the inconsistency of all pairs of different similarity indices. Then, we demonstrate that such disagreements often occur in practice when well-known clustering algorithms are applied to real datasets. Finally, we illustrate how an improper choice of a similarity index can affect the performance of production systems.

So, the question is: how to compare cluster similarity indices and choose the best one for a particular application? Ideally, we would want to choose an index for which good similarity scores translate to good real-world performance. However, opportunities to experimentally perform such a *validation of validation indices* are rare, typically expensive, and do not generalize to other applications. In contrast, we suggest a theoretical approach: we formally define properties that are desirable across various applications, discuss their importance, and formally analyze which similarity indices satisfy them (Section 4). This theoretical framework would allow practitioners to choose the best index

based on relevant properties for their applications. In Section 5, we advocate two indices that are expected to be suitable across various applications.

While many ideas discussed in the paper can be applied to all similarity indices, we also provide a more in-depth theoretical characterization of pair-counting ones (e.g., Rand and Jaccard), which gives an analytical background for further studies of pair-counting indices. We formally prove that among dozens of known indices, only two have all the properties except for being a distance: Correlation Coefficient and Sokal & Sneath's first index (Lei et al., 2017). Surprisingly, both indices are rarely used for cluster evaluation. The correlation coefficient has an additional advantage of being easily convertible to a distance measure via the arccosine function. The obtained index has all the properties except *constant baseline*, which is still satisfied asymptotically.

*Constant baseline* is a particular focus of the current research: this is one of the most important and non-trivial properties. Informally, a sensible index should not prefer one candidate partition over another just because it has too large or too small clusters. To the best of our knowledge, we are the first to develop a rigorous theoretical framework for analyzing this property. In this respect, our work improves over the previous (mostly empirical) research on constant baseline of particular indices (Albatineh et al., 2006; Lei et al., 2017; Strehl, 2002; Vinh et al., 2009; 2010), we refer to Appendix A for a detailed comparison to related research.

## 2 CLUSTER SIMILARITY INDICES

We assume that there is a set of elements $V$ with size $n = |V|$. A clustering is a partition of $V$ into disjoint subsets. Capital letters $A, B, C$ will be used to name the clusterings, and we will represent them as $A = \{A_1, \ldots, A_{k_A}\}$, where $A_i$ is the set of elements belonging to $i$-th cluster. If a pair of elements $v, w \in V$ lie in the same cluster in $A$, we refer to them as an *intra-cluster pair* of $A$, while *inter-cluster pair* will be used otherwise. The total number of pairs is denoted by $N = \binom{n}{2}$. The value that an index $I$ assigns to the similarity between partitions $A$ and $B$ will be denoted by $I(A, B)$. Let us now define some of the indices used throughout the paper, while a more comprehensive list, together with formal definitions, is given in Appendix B.1 and B.2.

**Pair-counting indices** consider clusterings to be similar if they agree on many pairs. Formally, let $\vec{A}$ be the $N$-dimensional vector indexed by the set of element-pairs, where the entry corresponding to $(v, w)$ equals 1 if $(v, w)$ is an intra-cluster pair and 0 otherwise. Further, let $M_{AB}$ be the $N \times 2$ matrix that results from concatenating the two (column-) vectors $\vec{A}$ and $\vec{B}$. Each row of $M_{AB}$ is either $11, 10, 01$, or $00$. Let the pair-counts $N_{11}, N_{10}, N_{01}, N_{00}$ denote the number of occurrences for each of these rows in $M_{AB}$.

**Definition 1.** *A pair-counting index is a similarity index that can be expressed as a function of the pair-counts* $N_{11}, N_{10}, N_{01}, N_{00}$.

Some popular pair-counting indices are *Rand* and *Jaccard*:

$$\text{R} = \frac{N_{11} + N_{00}}{N_{11} + N_{10} + N_{01} + N_{00}}, \quad \text{J} = \frac{N_{11}}{N_{11} + N_{10} + N_{01}}.$$

*Adjusted Rand* (AR) is a linear transformation of Rand ensuring that for a random $B$ we have $\text{AR}(A, B) = 0$ in expectation. A less widely used index is the Pearson *Correlation Coefficient* (CC) between the binary incidence vectors $\vec{A}$ and $\vec{B}$.[1] Another index, which we discuss further in more details, is the *Correlation Distance* $\text{CD}(A, B) := \frac{1}{\pi} \arccos \text{CC}(A, B)$. In Table 4, we formally define 27 known pair-counting indices and only mention ones of particular interest throughout the main text.

**Information-theoretic indices** consider clusterings similar if they share a lot of information, i.e., if little information is needed to transform one clustering into the other. Formally, let $H(A) := H(|A_1|/n, \ldots, |A_{k_A}|/n)$ be the Shannon entropy of the cluster-label distribution of $A$. Similarly, the joint entropy $H(A, B)$ is defined as the entropy of the distribution with probabilities $(p_{ij})_{i \in [k_A], j \in [k_B]}$, where $p_{ij} = |A_i \cap B_j|/n$. Then, the mutual information of two clusterings can be defined as

---

[1]Note that Spearman and Pearson correlation are equal when comparing binary vectors. Kendall rank correlation for binary vectors coincides with the Hubert index, which is linearly equivalent to Rand.

$M(A, B) = H(A) + H(B) - H(A, B)$. There are multiple ways of normalizing the mutual information, the most widely used ones are:

$$\text{NMI}(A, B) = \frac{M(A, B)}{(H(A) + H(B))/2}, \quad \text{NMI}_{\max}(A, B) = \frac{M(A, B)}{\max\{H(A), H(B)\}}.$$

NMI is known to be biased towards smaller clusters, and several modifications try to mitigate this bias: *Adjusted Mutual Information* (AMI) and *Standardized Mutual Information* (SMI) subtract the expected mutual information from $M(A, B)$ and normalize the obtained value (Vinh et al., 2009), while *Fair NMI* (FNMI) multiplies NMI by a penalty factor $e^{-|k_A - k_B|/k_A}$ (Amelio & Pizzuti, 2015).

## 3 MOTIVATING EXPERIMENTS

As discussed in Section 2, many cluster similarity indices are used by researchers and practitioners. A natural question is: *how to choose the best one?* Before trying to answer this question, it is important to understand whether the problem is relevant. Indeed, if the indices are very similar to each other and agree in most practical applications, then one can safely take any index. In this section, we demonstrate that this is not the case, and the proper choice matters.

First, we illustrate the inconsistency of all indices. We say that two indices $I_1$ and $I_2$ are inconsistent for a triplet of partitions $(A, B_1, B_2)$ if $I_1(A, B_1) > I_1(A, B_2)$ but $I_2(A, B_1) < I_2(A, B_2)$. We took 15 popular cluster similarity measures and constructed just four triplets such that each pair of indices is inconsistent for at least one triplet. One example is shown in Figure 1: for this simple example, about half of the indices prefer the left candidate, while the others prefer the right one. Other examples can be found in Appendix F.1.

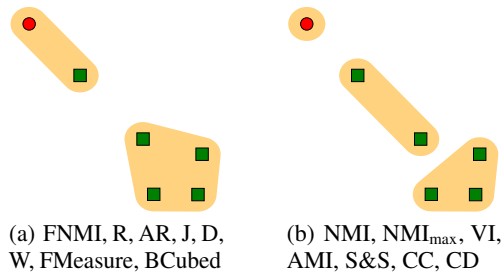

(a) FNMI, R, AR, J, D, W, FMeasure, BCubed

(b) NMI, NMI$_{\max}$, VI, AMI, S&S, CC, CD

Figure 1: Inconsistency of indices: shapes denote the reference partition, captions indicate indices favoring the candidate.

Thus, we see that the indices differ. But can this affect conclusions obtained in experiments on real data? To check that, we ran 8 well-known clustering algorithms (Scikit-learn, 2020) on 16 real-world datasets (GitHub, 2020). Each dataset, together with a pair of algorithms, gives a triplet of partitions $(A, B_1, B_2)$, where $A$ is a reference partition and $B_1, B_2$ are provided by two algorithms. For a given pair of indices and all such triplets, we look at whether the indices are consistent. Table 1 shows the relative inconsistency for several indices (the extended table together with a detailed description of the experimental setup and more analysis is given in Appendix F.2). The inconsistency rate is significant: e.g., popular measures Adjusted Rand and Variation if Information disagree in almost 40% of the cases, which is huge. Interestingly, the best agreeing indices are S&S and CC, which satisfy most of our properties, as shown in the next section. In contrast, the Variation of Information very often disagrees with other indices.

Table 1: Inconsistency of indices on real-world clustering datasets, %

|       | NMI  | VI   | AR   | S&S1 | CC   |
|-------|------|------|------|------|------|
| **NMI**  | –    | 40.3 | 15.7 | 20.1 | 18.5 |
| **VI**   |      | –    | 37.6 | 36.0 | 37.2 |
| **AR**   |      |      | –    | 11.7 | 8.3  |
| **S&S1** |      |      |      | –    | 3.6  |
| **CC**   |      |      |      |      | –    |

To show that the choice of similarity index may affect the final performance in a real production scenario, we conducted an experiment within a major news aggregator system. The system groups news articles to *events* and shows the list of most important events to users. For grouping, a clustering algorithm is used, and the quality of this algorithm affects the user experience: merging different clusters may lead to not showing an important event, while too much splitting may cause duplicate events. When comparing several candidate clustering algorithms, it is important to determine which one is the best for the system. Online experiments are expensive and can be used only for the best candidates. Thus, we need a tool for an offline comparison. For this purpose, we manually created a reference partition on a small fraction of news articles. We can use this partition to evaluate the

candidates. We performed such an offline comparison for two candidate algorithms and observed that different indices preferred different algorithms. Then, we launched an online user experiment and verified that one of the candidates is better for the system according to user preferences. Hence, it is important to be careful when choosing a similarity index for the offline comparison. See Appendix F.3 for more detailed description of this experiment and quantitative analysis.

## 4 ANALYSIS OF CLUSTER SIMILARITY INDICES

In this section, we motivate and formally define properties that are desirable for cluster similarity indices. We start with simple and intuitive ones that can be useful in some applications but not always necessary. Then, we discuss more complicated properties, ending with *constant baseline* that is extremely important but least trivial. In Tables 2 and 3, indices of particular interest are listed along with the properties satisfied. In Appendix C, we give the proofs for all entries of these tables.

For pair-counting indices we perform a more detailed analysis and define additional properties. For such indices, we interchangeably use the notation $I(A, B)$ and $I(N_{11}, N_{10}, N_{01}, N_{00})$.

Some of the indices have slight variants that are essentially the same. For example, the Hubert Index (Hubert, 1977) can be expressed as a linear transformation of the Rand index as $H(A, B) = 2R(A, B) - 1$. All the properties defined in this paper are invariant under linear transformations and interchanging $A$ and $B$. Hence, we define the following linear equivalence relation on similarity indices and check the properties for at most one representative of each equivalence class.

**Definition 2.** *Similarity indices $I_1$ and $I_2$ are* linearly equivalent *if there exists a nonconstant linear function $f$ such that either $I_1(A, B) = f(I_2(A, B))$ or $I_1(A, B) = f(I_2(B, A))$.*

This allows us to conveniently restrict to indices for which higher numerical values indicate higher similarity of partitions. Table 5 in the Appendix lists the equivalent indices.

### 4.1 PROPERTY 1: MAXIMAL AGREEMENT

The numerical value that an index assigns to a similarity must be easily interpretable. In particular, it should be easy to see whether the candidate clustering is maximally similar to (i.e., coincides with) the reference clustering. Formally, we require that $I(A, A) - c_{\max}$ is constant and either a strict upper bound for $I(A, B)$ for all $A \neq B$. The equivalence from Definition 2 allows us to assume that $I(A, A)$ is a maximum w.l.o.g. This property is easy to check, and it is satisfied by almost all indices, except for SMI and Wallace.

**Property 1′: Minimal agreement**   The maximal agreement property makes the upper range of the index interpretable. Similarly, having a numerical value for a low agreement would make the lower range interpretable. A minimal agreement is not well defined for general partitions: it is not clear which partition would be most dissimilar to a given one. However, referring to Lemma 1 in Appendix, pair-counting indices form a subclass of graph similarity indices. For a given graph $G = (V, E)$, it is clear that the graph most dissimilar to $G$ is its complement $G^C = (V, E^C)$. Comparing a graph to its complement would result in pair-counts $N_{11} = N_{00} = 0$ and $N_{10} + N_{01} = N$. This motivates the following definition:

**Definition 3.** *A pair-counting index $I$ has the* minimal agreement property *if there exists a constant $c_{\min}$ so that $I(N_{11}, N_{10}, N_{01}, N_{00}) \geq c_{\min}$ with equality if and only if $N_{11} = N_{00} = 0$.*

This property is satisfied by Rand, Correlation Coefficient, and Sokal&Sneath, while it is violated by Jaccard, Wallace, and Dice. Adjusted Rand does not have this property since substituting $N_{11} = N_{00} = 0$ gives the non-constant $\mathrm{AR}(0, N_{10}, N_{01}, 0) = -\frac{N_{10}N_{01}}{\frac{1}{2}N^2 - N_{10}N_{01}}$.

### 4.2 PROPERTY 3: SYMMETRY

Similarity is intuitively understood as a symmetric concept. Therefore, a good similarity index is expected to be symmetric, i.e., $I(A, B) = I(B, A)$ for all partitions $A, B$.[2] Tables 2 and 3

---

[2]In some applications, $A$ and $B$ may have different roles (e.g., reference and candidate partitions), and an asymmetric index may be suitable if there are different consequences of making false positives or false negatives.

Table 2: Requirements for general similarity indices

| | Max. agreement | Symmetry | Distance | Lin. complexity | Monotonicity | Const. baseline |
|---|---|---|---|---|---|---|
| **NMI** | ✓ | ✓ | ✗ | ✓ | ✓ | ✗ |
| **NMI$_{\max}$** | ✓ | ✓ | ✓ | ✓ | ✗ | ✗ |
| **FNMI** | ✓ | ✗ | ✗ | ✓ | ✗ | ✗ |
| **VI** | ✓ | ✓ | ✓ | ✓ | ✓ | ✗ |
| **SMI** | ✗ | ✓ | ✗ | ✗ | ✗ | ✓ |
| **FMeasure** | ✓ | ✓ | ✗ | ✓ | ✗ | ✗ |
| **BCubed** | ✓ | ✓ | ✗ | ✓ | ✓ | ✗ |
| **AMI** | ✓ | ✓ | ✗ | ✗ | ✓ | ✓ |

Table 3: Requirements for pair-counting indices[2]

| | Max. agreement | Min. agreement | Symmetry | Distance | Lin. complexity | Monotonicity | Strong monotonicity | Const. baseline | As. const. baseline | Type of bias |
|---|---|---|---|---|---|---|---|---|---|---|
| **R** | ✓ | ✓ | ✓ | ✓ | ✓ | ✓ | ✓ | ✗ | ✗ | ⤬ |
| **AR** | ✓ | ✗ | ✓ | ✗ | ✓ | ✓ | ✗ | ✓ | ✓ | |
| **J** | ✓ | ✗ | ✓ | ✓ | ✓ | ✓ | ✗ | ✗ | ✗ | ↘ |
| **W** | ✗ | ✗ | ✗ | ✗ | ✓ | ✗ | ✗ | ✗ | ✗ | ↘ |
| **D** | ✓ | ✗ | ✓ | ✗ | ✓ | ✓ | ✗ | ✗ | ✗ | ↙ |
| **CC** | ✓ | ✓ | ✓ | ✗ | ✓ | ✓ | ✓ | ✓ | ✓ | |
| **S&S1** | ✓ | ✓ | ✓ | ✗ | ✓ | ✓ | ✓ | ✓ | ✓ | |
| **CD** | ✓ | ✓ | ✓ | ✓ | ✓ | ✓ | ✓ | ✗ | ✓ | |

show that most indices are symmetric. The asymmetric ones are precision and recall (Wallace) and FNMI (Amelio & Pizzuti, 2015), which is a product of NMI and the asymmetric penalty factor.

### 4.3 PROPERTY 4: LINEAR COMPLEXITY

For clustering tasks on large datasets, running time is crucial, and algorithms with superlinear time can be infeasible. In these cases, a validation index with superlinear running time would be a significant bottleneck. Furthermore, computationally heavy indices also tend to be complicated and hard to interpret intuitively. We say that an index has *linear complexity* when its worst-case running time is $O(n)$. In Appendix C.2, we prove that any pair-counting index has $O(n)$ complexity. Many general indices have this property as well, except for SMI and AMI.

### 4.4 PROPERTY 4. DISTANCE

For some applications, a distance-interpretation of dissimilarity may be desirable: whenever $A$ is similar to $B$ and $B$ is similar to $C$, then $A$ should also be somewhat similar to $C$. For example, assume that we have the reference clustering that is an approximation of the ground truth (e.g., labeled by experts). In such situations, it may be reasonable to argue that the reference clustering is at most a distance $\varepsilon$ from the true clustering, so that the triangle inequality bounds the dissimilarity of the candidate clustering to the unknown true clustering.

A function $d$ is a distance metric if it satisfies three distance axioms: 1) symmetry ($d(A, B) = d(B, A)$); 2) positive-definiteness ($d(A, B) \geq 0$ with equality iff $A = B$); 3) the triangle inequality ($d(A, C) \leq d(A, B) + d(B, C)$). We say that $I$ is linearly transformable to a distance metric if there exists a linearly equivalent index that satisfies these three distance axioms. Note that all three axioms are invariant under re-scaling of $d$. We have already imposed the symmetry as a separate property, and the positive-definiteness is equivalent to the maximal agreement property. Therefore, whenever $I$ has these two properties, it satisfies the distance property iff $d(A, B) = c_{\max} - I(A, B)$ satisfies the triangle inequality, for $c_{\max}$ as defined in Section 4.1.

Examples of popular indices having this property are Variation of Information and the Mirkin metric. In Vinh et al. (2010), it is proved that when Mutual Information is normalized by the maximum of entropies, the resulting NMI is equivalent to a distance metric. A proof that the Jaccard index is equivalent to a distance is given in Kosub (2019). See Appendix C.1 for all the proofs.

**Correlation Distance** Among all the considered indices, there are two pair-counting ones having all the properties except for being a distance: Sokal&Sneath and Correlation Coefficient. However, the correlation coefficient can be transformed to a distance metric via a non-linear transformation. We

---

[2]All known pair-counting indices excluded from this table do not satisfy either constant baseline, symmetry, or maximal agreement.

define Correlation Distance (CD) as $\mathrm{CD}(A, B) := \frac{1}{\pi} \arccos \mathrm{CC}(A, B)$, where $\mathrm{CC}$ is the Pearson correlation coefficient and the factor $1/\pi$ scales the index to $[0, 1]$. To the best of our knowledge, this Correlation Distance has never before been used as a similarity index for comparing clusterings throughout the literature.

Let us point out that the Correlation Distance is indeed a distance. It follows from the fact that the correlation coefficient is obtained by first mapping the binary vectors $\vec{A}, \vec{B}$ to the unit sphere, and then taking their standard inner product. The arccosine of the inner product of two unit vectors corresponds to their angle, which is indeed a distance metric. A more detailed proof of this claim can be found in Appendix E.2. Further in this section, we show that the distance property of Correlation Distance is achieved at the cost of not having the exact constant baseline, which is still satisfied asymptotically.

### 4.5 PROPERTY 5: MONOTONICITY

When one clustering is changed such that it resembles the other clustering more, the similarity score ought to improve. Hence, we require an index to be monotone w.r.t. changes that increase the similarity. This can be formalized via the following definition.

**Definition 4.** *For clusterings $A$ and $B$, we say that $B'$ is an $A$-consistent improvement of $B$ iff $B \neq B'$ and all pairs of elements agreeing in $A$ and $B$ also agree in $A$ and $B'$.*

This leads to the following monotonicity property.

**Definition 5.** *An index $I$ satisfies the* monotonicity property *if for every two clusterings $A, B$ and any $B'$ that is an $A$-consistent improvement of $B$, it holds that $I(A, B') > I(A, B)$.*

To look at monotonicity from a different perspective, we define the following operations:

- **Perfect split**: $B'$ is a perfect split of $B$ (w.r.t. $A$) if $B'$ is obtained from $B$ by splitting a single cluster $B_1$ into two clusters $B'_1, B'_2$ such that no two elements of the same cluster of $A$ are in different parts of this split, i.e., for all $i$, $A_i \cap B_1$ is a subset of either $B'_1$ or $B'_2$.

- **Perfect merge**: We say that $B'$ is a perfect merge of $B$ (w.r.t. $A$) if there exists some $A_i$ and $B_1, B_2 \subset A_i$ such that $B'$ is obtained by merging $B_1, B_2$ into $B'_1$.

The following theorem gives an alternative definition of monotonicity and is proven in Appendix E.1.

**Theorem 1.** *$B'$ is an $A$-consistent improvement of $B$ iff $B'$ can be obtained from $B$ by a sequence of perfect splits and perfect merges.*

Note that this monotonicity is a stronger form of the first two constraints defined in (Amigó et al., 2009): *Cluster Homogeneity* is a weaker form of our monotonicity w.r.t. perfect splits, while *Cluster Equivalence* is equivalent to our monotonicity w.r.t. perfect merges.

Monotonicity is a critical property that should be satisfied by any sensible index. Surprisingly, not all indices satisfy this: we have found counterexamples that prove that SMI, FNMI, and Wallace do not have the monotonicity property. Furthermore, for NMI, whether monotonicity is satisfied depends on the normalization: the normalization by the average of the entropies has monotonicity, while the normalization by the maximum of the entropies does not.

**Property 5′. Strong monotonicity**    For pair-counting indices, we can define a stronger monotonicity property in terms of pair-counts.

**Definition 6.** *A pair-counting index $I$ satisfies* strong monotonicity *if it increases with $N_{11}, N_{00}$ and decreases with $N_{10}, N_{01}$.*

This property is stronger than monotonicity as it additionally allows for comparing similarities across different settings: we could compare the similarity between $A_1, B_1$ on $n_1$ elements with the similarity between $A_2, B_2$ on $n_2$ elements, even when $n_1 \neq n_2$. This ability to compare similarity scores across different numbers of elements is similar to the *Few data points* property of SMI (Romano et al., 2014) that allows its scale to have a similar interpretation across different settings.

We found several examples of indices that have Property 5 while not satisfying Property 5′. Jaccard and Dice indices are constant w.r.t. $N_{00}$, so they are not strongly monotone. A more interesting example is the Adjusted Rand index, which may become strictly larger if we only increase $N_{10}$.

### 4.6 PROPERTY 6. CONSTANT BASELINE.

This property is arguably the most significant: it is less intuitive than the other ones and may lead to unexpected consequences in practice. Informally, a good similarity index should not give a preference to a candidate clustering $B$ over another clustering $C$ just because $B$ has many or few clusters. This intuition can be formalized using random partitions: assume that we have some reference clustering $A$ and two random partitions $B$ and $C$. While intuitively both random guesses are equally bad approximations of $A$, it has been known throughout the literature (Albatineh et al., 2006; Romano et al., 2014; Vinh et al., 2010) that some indices tend to give higher scores for random guesses with a larger number of clusters. Ideally, we want the similarity value of a random candidate w.r.t. the reference partition to have a fixed expected value $c_{\text{base}}$ (independent of $A$). We formalize this in the following way. Let $S(B)$ denote the specification of the cluster sizes of the clustering $B$, i.e., $S(B) := [|B_1|, \dots, |B_{k_B}|]$, where $[\dots]$ denotes a multiset. For a cluster sizes specification $s$, let $\mathcal{C}(s)$ be the uniform distribution over clusterings $B$ with $S(B) = s$.

**Definition 7.** *An index $I$ satisfies the* constant baseline property *if there exists a constant $c_{base}$ so that* $\mathbf{E}_{B \sim \mathcal{C}(s)}[I(A, B)] = c_{base}$ *for any cluster-sizes specification $s$ and clustering $A$ with $1 < k_A < n$.*

Note that this property is symmetric since it does not matter whether we permute the labels of $A$ while keeping $B$ constant or vice versa. In the definition, we have excluded the cases where $A$ is a trivial clustering consisting of either 1 or $n$ clusters. Including them would cause problems for $s = S(A)$, as $\mathcal{C}(s)$ would be a constant distribution surely returning $A$ and any sensible index should have $I(A, A) \neq c_{\text{base}}$.

Constant baseline is extremely important in many practical applications: if an index violates this property, then its optimization may lead to undesirably biased results. For instance, if a biased index is used to choose the best algorithm among several candidates, then it is likely that the decision will be biased towards those who produce too large or too small clusters. This problem is often attributed to NMI (Romano et al., 2014; Vinh et al., 2010), but we found out that almost all indices suffer from it. The only indices that satisfy the constant baseline property are Adjusted Rand index, Correlation Coefficient, SMI, and AMI with $c_{\text{base}} = 0$ and Sokal&Sneath with $c_{\text{base}} = 1/2$. Interestingly, out of these five indices, three were *specifically designed* to satisfy this property, which made them less intuitive and resulted in other important properties being violated.

The only condition under which the constant baseline property can be safely ignored is knowing in advance *all cluster sizes*. In this case, bias towards particular cluster sizes would not affect decisions. However, we are not aware of any practical application where such an assumption can be made. Note that knowing only the number of clusters is insufficient. We illustrate this in Appendix D.4, where we also show that the bias of indices violating the constant baseline is easy to identify empirically.

**Property 6′: Asymptotic constant baseline** For pair-counting indices, a deeper analysis of the constant baseline property is possible. Let $m_A = N_{11} + N_{10}$, $m_B = N_{11} + N_{01}$ be the number of intra-cluster pairs of $A$ and $B$, respectively. Note that $m_A$ and $m_B$ are constant as $A$ is constant and $B \sim \mathcal{C}(s)$, so that its cluster-sizes are constant. Furthermore, the pair-counts $N_{10}, N_{01}, N_{00}$ are functions of $N, m_A, m_B, N_{11}$. Hence, to find the expected value of the index, we need to inspect it as a function of a single random variable $N_{11}$. For a random pair, the probability that it is an intra-cluster pair of both clusterings is $m_A m_B / N^2$, so the expected values of the pair-counts are

$$\overline{N_{11}} := \frac{m_A m_B}{N}, \ \overline{N_{10}} := m_A - \overline{N_{11}}, \ \overline{N_{01}} := m_B - \overline{N_{11}}, \ \overline{N_{00}} := N - m_A - m_B + \overline{N_{11}}. \quad (1)$$

We can use these values to define a weaker variant of constant baseline.

**Definition 8.** *A pair-counting index $I$ has the* asymptotic constant baseline property *if there exists a constant $c_{base}$ so that $I\left(\overline{N_{11}}, \overline{N_{10}}, \overline{N_{01}}, \overline{N_{00}}\right) = c_{base}$ for all $A$ with $1 < k_A < n$.*

In contrast to Definition 7, asymptotic constant baseline is very easy to verify: one just has to substitute the values from (1) to the index and check whether the obtained value is constant. Another important observation is that under mild assumptions $I(N_{11}, N_{10}, N_{01}, N_{00})$ converges in probability

to $I\left(\overline{N_{11}}, \overline{N_{10}}, \overline{N_{01}}, \overline{N_{00}}\right)$ as $n$ grows which justifies the usage of the name *asymptotic constant baseline*, see Appendix D.2 for more details.

Note that the non-linear transformation of Correlation Coefficient to Correlation Distance makes the latter one violate the constant baseline property. CD does, however, still have the asymptotic constant baseline at $1/2$ and we prove in Appendix E.3 that the expectation in Definition 7 is very close to this value. To the best of our knowledge, there does not exist a cluster similarity index that is a distance while having the exact constant baseline.

**Biases of cluster similarity indices**   Given the fact that there are so many biased indices, one may be interested in what kind of candidates they favor. While it is unclear how to formalize this concept for general validation indices, we can do this for pair-counting ones by analyzing them in terms of a single variable.

While there are previous attempts to characterize types of biases (Lei et al., 2017), they mostly rely on an analysis based on the number of clusters. However, we argue that the number of clusters is not a good measure of the granularity of a clustering. Instead, we show that the number of *inter-cluster pairs* should be analysed to determine the biases of pair-counting indices. We formally define and analyze two types of biases: *NPdec* and *NPinc*, where NP stands for Number of inter-cluster Pairs.

**Definition 9.** *Let $I$ be a pair-counting index and define $I^{(s)}(m_A, m_B) = I\left(\overline{N_{11}}, \overline{N_{10}}, \overline{N_{01}}, \overline{N_{00}}\right)$ for the expected pair-counts as defined in* (1). *We define the following biases:*

*(i)  $I$ suffers from* NPdec *bias if there are $m_A, m_B \in (0, N)$ such that $\frac{d}{dm_B} I^{(s)}(m_A, m_B) > 0$.*

*(ii)  $I$ suffers from* NPinc *bias if there are $m_A, m_B \in (0, N)$ such that $\frac{d}{dm_B} I^{(s)}(m_A, m_B) < 0$.*

Applying this definition to Jaccard $J^{(s)}(m_A, m_B) = \frac{m_A m_B}{N(m_A + m_B) - m_A m_B}$ and Rand $R^{(s)}(m_A, p_B) = 1 - (m_A + m_B)/N + 2m_A m_B/N^2$ immediately shows that Jaccard suffers from NPdec bias and Rand suffers from both biases, confirming the findings of Lei et al. (2017). Furthermore, the direction of the monotonicity for the bias of Rand is now determined by the condition $2m_A > N$ instead of the more complicated but equivalent condition on the quadratic entropy of $A$ that is given in Lei et al. (2017). Performing the same for Wallace and Dice shows that both suffer from NPdec bias. Note that an index satisfying the asymptotic constant baseline property will not have any of these biases as $I^{(s)}(m_A, m_B) = c_{\text{base}}$.

## 5   DISCUSSION AND CONCLUSION

At this point, we better understand the theoretical properties of cluster similarity indices, so it is time to answer the question: *which index is the best?* Unfortunately, there is no simple answer, but we can make an informed decision. In this section, we sum up what we have learned, argue that there are indices that are strictly better alternatives than some widely used ones, and give practical advice on how to choose a suitable index for a given application.

Among all properties discussed in this paper, *monotonicity* is the most crucial one. Violating this property is a fatal problem: such indices can prefer candidates which are strictly worse than others. Hence, we cannot advise using the well-known $\text{NMI}_{max}$, FMeasure, FNMI, and SMI indices.

The *constant baseline* property is much less trivial but is equally important: it addresses the problem of preferring some partitions only because they have small or large clusters. This property is essential unless you know *all cluster sizes*. Since we are not aware of practical applications where all cluster sizes are known, below we assume that this is not the case.[3] This requirement is satisfied by just a few indices, so we are only left with AMI, Adjusted Rand (AR), Correlation Coefficient (CC), and Sokal&Sneath (S&S). Additionally, Correlation Distance (CD) satisfies constant baseline asymptotically and deviations from the exact constant baseline are extremely small (see Section E.3).

Let us note that among the remaining indices, AR is strictly dominated by CC and S&S since it does not have the minimum agreement and strong monotonicity. Also, similarly to AMI, AR is specifically

---

[3]However, in applications where such an assumption holds, it can be reasonable to use, e.g., BCubed, Variation of Information, and NMI.

created to have a constant baseline, which made this index more complex and less intuitive than other pair-counting indices. Hence, we are only left with four indices: AMI, S&S, CC, and CD.

According to their theoretical properties, all these indices are good, and any of them can be chosen. Figure 2 illustrates how a final decision can be made. First, one can decide whether the distance property is needed. For example, suppose one wants to cluster the algorithms by comparing the partitions provided by them. In that case, the metric property of a similarity index allows the use of metric clustering algorithms. In this case, a distance property is desirable, and CD is the best choice: it has all properties except for the exact constant baseline, which is still satisfied asymptotically. Next, it is important to decide whether computation time is of the essence. Linear computation time is essential for large-scale applications.

For instance, assume that there is a production system that groups news articles or user photos. There is a candidate algorithm, and we want to compare it with the currently used one to avoid major changes. In this case, we have to compare huge partitions, and time is of the essence. Another example is multiple comparisons: choosing the best algorithm among many candidates (differing, e.g., by a parameter value). If this is the case, then AMI is not a proper choice, and one has to choose between CC and S&S. Otherwise, all three indices are suitable according to our formal constraints.

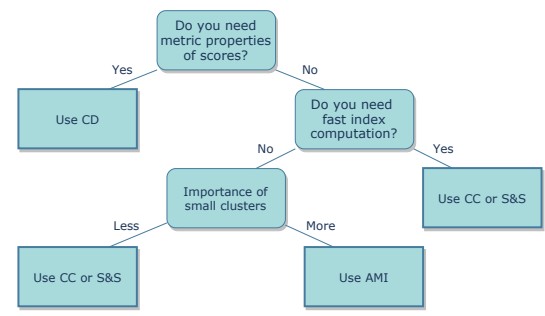

Figure 2: Example of how one can make a decision among good cluster similarity indices.

Let us discuss an (informal) criterion that may help to choose between AMI and pair-counting alternatives. Different indices may favor a different balance between errors in small and large clusters. In particular, all pair-counting indices give larger weights to errors in large clusters: misclassifying one element in a cluster of size $k$ costs $k-1$ incorrect pairs. It is known (empirically) that information-theoretic indices do not have this property and give a higher weight to small clusters (Amigó et al., 2009).[4] Amigó et al. (2009) argue that for their particular application (text clustering), it is desirable not to give a higher weight to large clusters. In contrast, there are applications where the opposite may hold. For instance, consider a system that groups user photos based on identity and shows these clusters to a user as a ranked list. In this case, a user is likely to investigate the largest clusters consisting of known people and would rarely spot an error in a small cluster. The same applies to any system that ranks the clusters, e.g., to news aggregators. Based on what is desirable for a particular application, one can choose between AMI and pair-counting CC and S&S.

The final decision between CC and S&S is hard to make since they are equally good in terms of their theoretical properties. Interestingly, although some works (Choi et al., 2010; Lei et al., 2017) list Pearson correlation as a cluster similarity index, it has not received attention that our results suggest it deserves, similarly to S&S. First, both indices are interpretable. CC is a correlation between the two incidence vectors, which is a very natural concept. S&S is the average of precision and recall (for binary classification of pairs) plus their inverted counterparts, which can also be intuitively understood. Also, CC and S&S usually agree in practice: in Tables 1 and 6 we can see that they have the largest agreement. Hence, one can take any of these indices. Another option would be to check whether there are situations where these indices disagree and, if this happens, perform an experiment similar to what we did in Section 3 for news aggregation.

Finally, while some properties listed in Tables 2 and 3 are not mentioned in the discussion above, they can be important for particular applications. For instance, maximum and minimum agreements are useful for interpretability, but they can also be essential if some operations are performed over the index values: e.g., averaging the scores of different algorithms. Symmetry can be necessary if there is no "gold standard" partition, but algorithms are compared only to each other.

---

[4]This is an interesting aspect that has not received much attention in our research since we believe that the desired balance between large and small clusters may differ per application and we are not aware of a proper formalization of this "level of balance" in a general form.

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

## A FURTHER RELATED WORK

Several attempts to the comparative analysis of cluster similarity indices have been made in the literature, both in machine learning and complex networks communities. In particular, the problem of indices favoring clusterings with smaller or larger clusters has been identified (Albatineh et al., 2006; Lei et al., 2017; Vinh et al., 2009; 2010). The most popular approach to resolving the bias of an index is to subtract its expected value and normalize the resulting quantity to obtain an index that satisfies the maximum agreement property. This approach has let to 'adjusted' indices such as AR (Hubert & Arabie, 1985) and AMI (Vinh et al., 2009). In Albatineh et al. (2006), the family of pair-counting indices $\mathcal{L}$ is introduced for which adjusted forms can be computed easily. This family corresponds to the set of all pair-counting indices that are linear functions of $N_{11}$ for fixed $N_{11} + N_{10}, N_{11} + N_{01}$. A generalization of information-theoretic indices by Tsallis $q$-entropy is given in Romano et al. (2016) and is shown to correspond to pair-counting indices for $q = 2$. Formulas are provided for adjusting these generalized indices for chance.

A disadvantage of this adjustment scheme is that an index can be normalized in many ways, while it is difficult to grasp the differences between these normalizations intuitively. For example, three

variants of AMI have been introduced (Vinh et al., 2009), and we show that normalization by the maximum entropies results in an index that fails monotonicity. Romano et al. (2014) go one step further by standardizing mutual information, while Amelio & Pizzuti (2015) multiply NMI with a penalty factor that decreases with the difference in the number of clusters.

In summary, all these works take a popular biased index and 'patch' it to get rid of this bias. This approach has two disadvantages: firstly, these patches often introduce new problems (e.g., FNMI and SMI fail monotonicity), and secondly, the resulting index is usually less interpretable than the original. We have taken a different approach in our work: instead of patching existing indices, we analyze previously introduced indices to see whether they satisfy more properties. Our analysis shows that ARI is dominated by Pearson correlation, which was introduced more than 100 years before ARI. Therefore, there was no need to construct ARI from Rand in the first place.

In Lei et al. (2017), the biases of pair-counting indices are characterized. They define these biases as a preference towards either few or many clusters. They prove that the direction of Rand's bias depends on the Havrda-Charvat entropy of the reference clustering. In the present work, we show that the number of clusters is not an adequate quantity for expressing these biases. We introduce methods to easily analyze the bias of any pair-counting index and simplify the condition for the direction of Rand's bias to $m_A < N/2$.

A paper closely related to the current research (Amigó et al., 2009) formulates several constraints (axioms) for cluster similarity indices. Their *cluster homogeneity* is a weaker analog of our monotonicity w.r.t. perfect splits while their *cluster equivalence* is equivalent to our monotonicity w.r.t. perfect merges. The third *rag bag* constraint is motivated by a subjective claim that "introducing disorder into a disordered cluster is less harmful than introducing disorder into a clean cluster". While this is important for their particular application (text clustering), we found no other work that deemed this constraint necessary; hence, we disregarded this constraint in the current research. The last constraint by Amigó et al. (2009) concerns the balance between making errors in large and small clusters. Though this is an interesting aspect that has not received much attention in our research, this constraint poses a particular balance while we believe that the desired balance may differ per application. Hence, this property seems to be non-binary and we are not aware of a proper formalization of this "level of balance" in a general form. Hence, we do not include this in our list of formal properties. The most principal difference of our work compared to Amigó et al. (2009) is the constant baseline which was not analyzed in their work. We find this property extremely important while it is failed by most of the widely used indices including their BCubed. To conclude, our research gives a more comprehensive list of constraints and focuses on those that are desirable in a wide range of applications. We also cover all similarity indices often used in the literature and give formal proofs for all index-property combinations.

A property similar to our monotonicity property is also given in Meilă (2007), where the similarity between clusterings $A$ and $B$ is upper-bounded by the similarity between $A$ and $A \otimes B$ (as defined in Section C.4). One can show that this property is implied by our monotonicity but not vice versa, i.e., the variant proposed by Meilă (2007) is weaker. Our analysis of monotonicity generalizes and unifies previous approaches to this problem, see Theorem 1 that relates consistent improvements to perfect splits and merges.

While we focus on *external* cluster similarity indices that compare a candidate partition with a reference one, there are also *internal* similarity measures that estimate the quality of partitions with respect to internal structure of data (e.g., Silhouette, Hubert-Gamma, Dunn, and many other indices). Kleinberg (2002) used an axiomatic approach for internal measures and proved an impossibility theorem: there are three simple and natural constraints such that no internal clustering measure can satisfy all of them. More work in this direction can be found in, e.g., Ben-David & Ackerman (2008). In network analysis, internal measures compare a candidate partition with the underlying graph structure. They quantify how well a community structure (given by a partition) fits the graph and are often referred to as goodness or quality measures. The most well-known example is *modularity* (Newman & Girvan, 2004). Axioms that these measures ought to satisfy are given in (Ben-David & Ackerman, 2009; Van Laarhoven & Marchiori, 2014). Note that all pair-counting indices discussed in this paper can also be used for graph-partition similarity, as we discuss in Section B.3.

# B  CLUSTER SIMILARITY INDICES

## B.1  GENERAL INDICES

Here we give the definitions of the indices listed in Table 2. We define the contingency variables as $n_{ij} = |A_i \cap B_j|$. We note that all indices discussed in this paper can be expressed as functions of these contingency variables.

The *F-Measure* is defined as the harmonic mean of recall and precision. Recall is defined as

$$r(A, B) = \frac{1}{n} \sum_{i=1}^{k_A} \max_{j \in [k_B]} \{n_{ij}\},$$

and precision is its symmetric counterpart $r(B, A)$.

In (Amigó et al., 2009), recall is redefined as

$$r'(A, B) = \frac{1}{n} \sum_{i=1}^{k_A} \frac{1}{|A_i|} \sum_{j=1}^{k_B} n_{ij}^2,$$

and *BCubed* is defined as the harmonic mean of $r'(A, B)$ and $r'(B, A)$.

The remainder of the indices are information-theoretic and require some additional definitions. Let $p_1, \ldots, p_\ell$ be a discrete distribution (i.e., all values are nonnegative and sum to 1). The Shannon entropy is then defined as

$$H(p_1, \ldots, p_\ell) := -\sum_{i=1}^{\ell} p_i \log(p_i).$$

The entropy of a clustering is defined as the entropy of the cluster-label distribution of a random item, i.e.,

$$H(A) := H(|A_1|/n, \ldots, |A_{k_A}|/n),$$

and similarly for $H(B)$. The joint entropy $H(A, B)$ is then defined as the entropy of the distribution with probabilities $(p_{ij})_{i \in [k_A], j \in [k_B]}$, where $p_{ij} = n_{ij}/n$.

*Variation of Information* (Meilă, 2007) is defined as

$$\mathrm{VI}(A, B) = 2H(A, B) - H(A) - H(B).$$

Mutual information is defined as

$$M(A, B) = H(A) + H(B) - H(A, B).$$

The mutual information between $A$ and $B$ is upper-bounded by $H(A)$ and $H(B)$, which gives multiple possibilities to normalize the mutual information. In this paper, we discuss two normalizations: normalization by the average of the entropies $\frac{1}{2}(H(A) + H(B))$, and normalization by the maximum of entropies $\max\{H(A), H(B)\}$. We will refer to the corresponding indices as NMI and $\mathrm{NMI}_{\max}$, respectively:

$$\mathrm{NMI}(A, B) = \frac{M(A, B)}{(H(A) + H(B))/2},$$

$$\mathrm{NMI}_{\max}(A, B) = \frac{M(A, B)}{\max\{H(A), H(B)\}}.$$

*Fair NMI* is a variant of NMI that includes a factor that penalizes large differences in the number of clusters (Amelio & Pizzuti, 2015). It is given by

$$\mathrm{FNMI}(A, B) = e^{-|k_A - k_B|/k_A} \mathrm{NMI}(A, B).$$

In this definition, NMI may be normalized in various ways. We note that a different normalization would not result in more properties being satisfied.

*Adjusted Mutual Information* addresses for the bias of NMI by subtracting the expected mutual informationVinh et al. (2009). It is given by

$$\text{AMI}(A, B) = \frac{M(A, B) - \mathbf{E}_{B' \sim \mathcal{C}(S(B))}[M(A, B')]}{\sqrt{H(A) \cdot H(B)} - \mathbf{E}_{B' \sim \mathcal{C}(S(B))}[M(A, B')]}.$$

Here, a normalization by the geometric mean of the entropies is used, while other normalizations are also used (Vinh et al., 2009).

*Standardized Mutual Information* standardizes the mutual information w.r.t. random permutations of the items (Romano et al., 2014), i.e.,

$$\text{SMI}(A, B) = \frac{M(A, B) - \mathbf{E}_{B' \sim \mathcal{C}(S(B))}(M(A, B'))}{\sigma_{B' \sim \mathcal{C}(S(B))}(M(A, B'))},$$

where $\sigma$ denotes the standard deviation. Calculating the expected value and standard deviation of the mutual information is nontrivial and requires significantly more computation power than other indices. For this, we refer to the original paper (Romano et al., 2014). Note that this index is symmetric since it does not matter whether we keep $A$ constant while randomly permuting $B$ or keep $B$ constant while randomly permuting $A$.

### B.2 PAIR-COUNTING INDICES AND THEIR EQUIVALENCES

Pair-counting similarity indices are defined in Table 4. Table 5 lists linearly equivalent indices (see Definition 2). Note that our linear equivalence differs from the less restrictive monotonous equivalence given in (Batagelj & Bren, 1995). In the current work, we have to restrict to linear equivalence as the constant baseline property is not invariant to non-linear transformations.

### B.3 DEFINING THE SUBCLASS OF PAIR-COUNTING INDICES

From Definition 1 of the main text, it follows that a pair-counting index is a function of two binary vectors $\vec{A}, \vec{B}$ of length $N$. Note that this binary-vector representation has some redundancy: whenever $u, v$ and $v, w$ form intra-cluster pairs, we know that $u, w$ must also be an intra-cluster pair. Hence, not every binary vector of length $N$ represents a clustering. The class of $N$-dimensional binary vectors is, however, isomorphic to the class of undirected graphs on $n$ vertices. Therefore, pair-counting indices are also able to measure the similarity between graphs. For example, for an undirected graph $G = (V, E)$, one can consider its incidence vector $\vec{G} = (\mathbf{1}\{\{v, w\} \in E\})_{v, w \in V}$. Hence, pair-counting indices can be used to measure the similarity between two graphs or between a graph and a clustering. So, one may see a connection between graph and cluster similarity indices. For example, the Mirkin metric is a pair-counting index that coincides with the Hamming distance between the edge-sets of two graphs (Donnat & Holmes, 2018). Another example is the Jaccard graph distance, which turns out to be more appropriate for comparing sparse graphs (Donnat & Holmes, 2018). Thus, all pair-counting indices and their properties discussed in the current paper can also be applied to graph-graph and graph-partition similarities.

In this section, we show that the subclass of pair-counting similarity indices can be uniquely defined by the property of being pair-symmetric.

For two graphs $G_1$ and $G_2$ let $M_{G_1 G_2}$ denote the $N \times 2$ matrix that is obtained by concatenating their adjacency vectors. Let us write $I_M^{(G)}(M_{G_1 G_2})$ for the similarity between two graphs $G_1, G_2$ according to some graph similarity index $I^{(G)}$. We will now characterize all pair-counting similarity indices as a subclass of the class of similarity indices between undirected graphs.

**Definition 10.** *We define a graph similarity index $I_M^{(G)}(M_{G_1 G_2})$ to be* pair-symmetric *if interchanging two rows of $M_{G_1, G_2}$ leaves the index unchanged.*

We give the following result.

**Lemma 1.** *The class of pair-symmetric graph similarity indices coincides with the class of pair-counting cluster similarity indices.*

---

[5]Throughout the literature, the Mirkin metric is defined as $2(N_{10} + N_{01})$, but we use this variant as it satisfies the scale-invariance.

Table 4: A selection of pair-counting indices. Most of these indices are taken from Lei et al. (2017).

| Index (Abbreviation) | Expression |
|---|---|
| Rand ($R$) | $\frac{N_{11}+N_{00}}{N_{11}+N_{10}+N_{01}+N_{00}}$ |
| Adjusted Rand ($AR$) | $\frac{N_{11}-\frac{(N_{11}+N_{10})(N_{11}+N_{01})}{N_{11}+N_{10}+N_{01}+N_{00}}}{\frac{(N_{11}+N_{10})+(N_{11}+N_{01})}{2}-\frac{(N_{11}+N_{10})(N_{11}+N_{01})}{N_{11}+N_{10}+N_{01}+N_{00}}}$ |
| Jaccard ($J$) | $\frac{N_{11}}{N_{11}+N_{10}+N_{01}}$ |
| Jaccard Distance ($JD$) | $\frac{N_{10}+N_{01}}{N_{11}+N_{10}+N_{01}}$ |
| Wallace1 ($W$) | $\frac{N_{11}}{N_{11}+N_{10}}$ |
| Wallace2 | $\frac{N_{11}}{N_{11}+N_{01}}$ |
| Dice | $\frac{2N_{11}}{2N_{11}+N_{10}+N_{01}}$ |
| Correlation Coefficient ($CC$) | $\frac{N_{11}N_{00}-N_{10}N_{01}}{\sqrt{(N_{11}+N_{10})(N_{11}+N_{01})(N_{00}+N_{10})(N_{00}+N_{01})}}$ |
| Correlation Distance ($CD$) | $\frac{1}{\pi}\arccos\left(\frac{N_{11}N_{00}-N_{10}N_{01}}{\sqrt{(N_{11}+N_{10})(N_{11}+N_{01})(N_{00}+N_{10})(N_{00}+N_{01})}}\right)$ |
| Sokal&Sneath-I ($S\&S_1$) | $\frac{1}{4}\left(\frac{N_{11}}{N_{11}+N_{10}}+\frac{N_{11}}{N_{11}+N_{01}}+\frac{N_{00}}{N_{00}+N_{10}}+\frac{N_{00}}{N_{00}+N_{01}}\right)$ |
| Minkowski | $\sqrt{\frac{N_{10}+N_{01}}{N_{11}+N_{10}}}$ |
| Hubert ($H$) | $\frac{N_{11}+N_{00}-N_{10}-N_{01}}{N_{11}+N_{10}+N_{01}+N_{00}}$ |
| Folkes&Mallow | $\frac{N_{11}}{\sqrt{(N_{11}+N_{10})(N_{11}+N_{01})}}$ |
| Sokal&Sneath-II | $\frac{\frac{1}{2}N_{11}}{\frac{1}{2}N_{11}+N_{10}+N_{01}}$ |
| Normalized Mirkin[5] | $\frac{N_{10}+N_{01}}{N_{11}+N_{10}+N_{01}+N_{00}}$ |
| Kulczynski | $\frac{1}{2}\left(\frac{N_{11}}{N_{11}+N_{10}}+\frac{N_{11}}{N_{11}+N_{01}}\right)$ |
| McConnaughey | $\frac{N_{11}^2-N_{10}N_{01}}{(N_{11}+N_{10})(N_{11}+N_{01})}$ |
| Yule | $\frac{N_{11}N_{00}-N_{10}N_{01}}{N_{11}N_{10}+N_{01}N_{00}}$ |
| Baulieu-I | $\frac{(N_{11}+N_{10}+N_{01}+N_{00})(N_{11}+N_{00})+(N_{10}-N_{01})^2}{(N_{11}+N_{10}+N_{01}+N_{00})^2}$ |
| Russell&Rao | $\frac{N_{11}}{N_{11}+N_{10}+N_{01}+N_{00}}$ |
| Fager&McGowan | $\frac{N_{11}}{\sqrt{(N_{11}+N_{10})(N_{11}+N_{01})}}-\frac{1}{2\sqrt{N_{11}+N_{10}}}$ |
| Peirce | $\frac{N_{11}N_{00}-N_{10}N_{01}}{(N_{11}+N_{01})(N_{00}+N_{10})}$ |
| Baulieu-II | $\frac{N_{11}N_{00}-N_{10}N_{01}}{(N_{11}+N_{10}+N_{01}+N_{00})^2}$ |
| Sokal&Sneath-III | $\frac{N_{11}N_{00}}{\sqrt{(N_{11}+N_{10})(N_{11}+N_{01})(N_{00}+N_{10})(N_{00}+N_{01})}}$ |
| Gower&Legendre | $\frac{N_{11}+N_{00}}{N_{11}+\frac{1}{2}(N_{10}+N_{01})+N_{00}}$ |
| Rogers&Tanimoto | $\frac{N_{11}+N_{00}}{N_{11}+2(N_{10}+N_{01})+N_{00}}$ |
| Goodman&Kruskal | $\frac{N_{11}N_{00}-N_{10}N_{01}}{N_{11}N_{00}+N_{10}N_{01}}$ |

*Proof.* A matrix is an ordered list of its rows. An unordered list is a multiset. Hence, when we disregard the ordering of the matrix $M_{AB}$, we get a multiset of the rows. This multiset contains at most four distinct elements, each corresponding to one of the pair-counts. Therefore, each $I_M^{(G)}(M_{AB})$ that is symmetric w.r.t. interchanging rows is equivalently a function of the pair-counts of $A$ and $B$. □

Table 5: Equivalent pair-counting indices

| Representative Index | Equivalent indices |
|---|---|
| Rand | Normalized Mirkin Metric, Hubert |
| Jaccard | Jaccard Distance |
| Wallace1 | Wallace2 |
| Kulczynski | McConnaughey |

## C  CHECKING PROPERTIES FOR INDICES

In this section, we check all non-trivial properties for all indices. The properties of symmetry, maximal/minimal agreement and asymptotic constant baseline can trivially be tested by simply checking $I(B, A) = I(A, B)$, $I(A, A) = c_{\max}$, $I(0, N_{10}, N_{01}, 0) = c_{\min}$ and $I^{(p)}(p_A p_B, p_A, p_B) = c_{\text{base}}$ respectively.

### C.1  DISTANCE

#### C.1.1  POSITIVE CASES

**NMI and VI.**  In (Vinh et al., 2010) it is proven that for max-normalization $1 - \text{NMI}$ is a distance, while in (Meilă, 2007) it is proven that VI is a distance.

**Rand.**  The Mirkin metric $1 - R$ corresponds to a rescaled version of the size of the symmetric difference between the sets of intra-cluster pairs. The symmetric difference is known to be a distance metric.

**Jaccard.**  In Kosub (2019), it is proven that the Jaccard distance $1 - J$ is indeed a distance.

**Correlation Distance.**  In Theorem E.2 it is proven that Correlation Distance is indeed a distance.

#### C.1.2  NEGATIVE CASES

To prove that an index that satisfies symmetry and maximal agreement is not linearly transformable to a distance metric, we only need to disprove the triangle inequality for one instance of its equivalence class that is nonnegative and equals zero for maximal agreement.

**FNMI and Wallace.**  These indices cannot be transformed to distances as they are not symmetric.

**SMI.**  SMI does not satisfy the maximal agreement property (Romano et al., 2014), so it cannot be transformed to a metric.

**FMeasure and BCubed.**  We will use a simple counter-example, where $|V| = 3, k_A = 1, k_B = 2, k_C = 3$. Let us denote the FMeasure and BCubed by $FM, BC$ respectively. We get

$$1 - \text{FM}(A, C) = 1 - 0.5 > (1 - 0.8) + (1 - 0.8) = (1 - \text{FM}(A, B)) + (1 - \text{FM}(B, C))$$

and

$$1 - \text{BC}(A, C) = 1 - 0.5 > (1 - 0.71) + (1 - 0.8) \approx (1 - \text{BC}(A, B)) + (1 - \text{BC}(B, C)),$$

so that both indices violate the triangle inequality in this case.

**Adjusted Rand, Dice, Correlation Coefficient, Sokal&Sneath and AMI.**  For these indices, we use the following counter-example: Let $A = \{\{0, 1\}, \{2\}, \{3\}\}, B = \{\{0, 1\}, \{2, 3\}\}, C = \{\{0\}, \{1\}, \{2, 3\}\}$. Then $p_{AB} = p_{BC} = 1/6$ and $p_{AC} = 0$ while $p_A = p_C = 1/6$ and $p_B = 1/3$. By substituting these variables, one can see that

$$1 - I^{(p)}(p_{AC}, p_A, p_C) > (1 - I^{(p)}(p_{AB}, p_A, p_B)) + (1 - I^{(p)}(p_{BC}, p_B, p_C)),$$

holds for each of these indices, contradicting the triangle inequality. The same $A, B$ and $C$ also form a counter-example for AMI.

## C.2 LINEAR COMPLEXITY

We will frequently make use of the following lemma:

**Lemma 2.** *The nonzero values of $n_{ij}$ can be computed in $O(n)$.*

*Proof.* We will store these nonzero values in a hash-table that maps the pairs $(i, j)$ to their value $n_{ij}$. These values are obtained by iterating through all items and incrementing the corresponding value of $n_{ij}$. For hash-tables, searches and insertions are known to have amortized complexity complexity $O(1)$, meaning that any sequence of $n$ such actions has worst-case running time of $O(n)$, from which the result follows. □

### C.2.1 POSITIVE CASES

**NMI, FNMI and VI.** Given the positive values of $n_{ij}$, it is clear that the joint and marginal entropy values can be computed in $O(n)$. From these values, the indices can be computed in constant time, leading to a worst-case running time of $O(n)$.

**FMeasure and BCubed.** Note that in the expressions of recall and precision as defined by these indices, only the positive values of $n_{ij}$ contribute. Furthermore, all of the variables $a_i, b_j$ and $n_{ij}$ appear at most once, so that these can indeed be computed in $O(n)$.

**Pair-counting indices.** Note that $N_{11} = \sum_{n_{ij}>0} \binom{n_{ij}}{2}$ can obviously be computed in $O(n)$. Similarly, $m_A = \sum_{i=1}^{k_A} \binom{a_i}{2}$ and $m_B$ can be computed in $O(k_A), O(k_B)$ respectively. The other pair-counts are then obtained by $N_{10} = m_A - N_{11}$, $N_{01} = m_B - N_{11}$ and $N_{00} = N - m_A - m_B + N_{11}$.

### C.2.2 AMI AND SMI.

Both of these require the computation of the expected mutual information. It has been known Romano et al. (2016) that this has a worst-case running time of $O(n \cdot \max\{k_A, k_B\})$ while $\max\{k_A, k_A\}$ can be $O(n)$.

## C.3 STRONG MONOTONICITY

### C.3.1 POSITIVE CASES

**Correlation Coefficient.** This index has the property that inverting one of the binary vectors results in the index flipping sign. Furthermore, the index is symmetric. Therefore, we only need to prove that this index is increasing in $N_{11}$. We take the derivative and omit the constant factor $((N_{00} + N_{10})(N_{00} + N_{01}))^{-\frac{1}{2}}$:

$$\frac{N_{00}}{\sqrt{(N_{11} + N_{10})(N_{11} + N_{01})}} - \frac{(N_{11}N_{00} - N_{10}N_{01}) \cdot \frac{1}{2}(2N_{11} + N_{10} + N_{01})}{[(N_{11} + N_{10})(N_{11} + N_{01})]^{1.5}}$$
$$= \frac{\frac{1}{2}N_{11}N_{00}(N_{10} + N_{01}) + N_{00}N_{10}N_{01}}{[(N_{11} + N_{10})(N_{11} + N_{01})]^{1.5}} + \frac{\frac{1}{2}N_{10}N_{01}(2N_{11} + N_{10} + N_{01})}{[(N_{11} + N_{10})(N_{11} + N_{01})]^{1.5}} > 0.$$

**Correlation Distance.** The correlation distance satisfies strong monotonicity as it is a monotone transformation of the correlation coefficient, which meets the property.

**Sokal&Sneath.** All four fractions are nondecreasing in $N_{11}, N_{00}$ and nonincreasing in $N_{10}, N_{01}$ while for each of the variables there is one fraction that satisfies the monotonicity strictly so that the index is strongly monotonous.

**Rand Index.** For the Rand index, it can be easily seen from the form of the index that it is increasing in $N_{11}, N_{00}$ and decreasing in $N_{10}, N_{01}$ so that it meets the property.

### C.3.2 NEGATIVE CASES

**Jaccard, Wallace, Dice.** All these three indices are constant w.r.t. $N_{00}$. Therefore, these indices do not satisfy strong monotonicity.

**Adjusted Rand.** It holds that

$$AR(1, 2, 1, 0) < AR(1, 3, 1, 0),$$

so that the index does not meet the strong monotonicity property.

## C.4 MONOTONICITY

### C.4.1 POSITIVE CASES

**Rand, Correlation Coefficient, Sokal&Sneath, Correlation Distance.** Strong monotonicity implies monotonicity. Therefore, these pair-counting indices satisfy the monotonicity property.

**Jaccard and Dice.** It can be easily seen that these indices are increasing in $N_{11}$ while decreasing in $N_{10}, N_{01}$. For $N_{00}$, we note that whenever $N_{00}$ gets increased, either $N_{10}$ or $N_{01}$ must decrease, resulting in an increase of the index. Therefore, these indices satisfy monotonicity.

**Adjusted Rand.** Note that for $b, b + d > 0$, it holds that

$$\frac{a + c}{b + d} > \frac{a}{b} \Leftrightarrow c > \frac{ad}{b}. \tag{2}$$

For Adjusted Rand, we have

$$a = N_{11} - \frac{1}{N}(N_{11} + N_{10})(N_{11} + N_{01}), \quad b = a + \frac{1}{2}(N_{10} + N_{01}).$$

Because of this, when we increment either $N_{11}$ or $N_{00}$ while decrementing either $N_{10}$ or $N_{01}$, we get $d = c - \frac{1}{2}$. Hence, we need to prove $c > a(c - \frac{1}{2})/b$, or, equivalently

$$c > -\frac{a}{2(b - a)} = \frac{\frac{1}{N}(N_{11} + N_{10})(N_{11} + N_{01}) - N_{11}}{N_{10} + N_{01}}.$$

For simplicity we rewrite this to

$$c + \frac{p_{AB} - p_A p_B}{p_A + p_B - 2p_{AB}} > 0,$$

where $p_A = \frac{1}{N}(N_{11} + N_{10}) \in (0, 1)$ and $p_B = \frac{1}{N}(N_{11} + N_{01}) \in (0, 1)$. If we increment $N_{00}$ while decrementing either $N_{10}$ or $N_{01}$, then

$$c \in \left\{ \frac{1}{N}(N_{11} + N_{10}), \frac{1}{N}(N_{11} + N_{01}) \right\} = \{p_A, p_B\}.$$

The symmetry of AR allows us to w.l.o.g. assume that $c = p_A$. We write

$$p_A + \frac{p_{AB} - p_A p_B}{p_A + p_B - 2p_{AB}} = \frac{p_A^2 + (1 - 2p_A)p_{AB}}{p_A + p_B - 2p_{AB}}.$$

When $p_A \leq \frac{1}{2}$, then this is clearly positive. For the case $p_A > \frac{1}{2}$, we bound $p_{AB} \leq p_A$ and bound the numerator by

$$p_A^2 + (1 - 2p_A)p_A = (1 - p_A)p_A > 0.$$

This proves the monotonicity for increasing $N_{00}$. When incrementing $N_{11}$ while decrementing either $N_{10}$ or $N_{01}$, we get $c \in \{1 - p_A, 1 - p_B\}$. Again, we assume w.l.o.g. that $c = 1 - p_A$ and write

$$1 - p_A + \frac{p_{AB} - p_A p_B}{p_A + p_B - 2p_{AB}} = \frac{p_A(1 - p_A) + (1 - 2p_A)(p_B - p_{AB})}{p_A + p_B - 2p_{AB}}.$$

This is clearly positive whenever $p_A \leq \frac{1}{2}$. When $p_A > \frac{1}{2}$, we bound $p_{AB} \geq p_A + p_B - 1$ and rewrite the numerator as

$$p_A(1 - p_A) + (1 - 2p_A)(p_A - 1) = (1 - p_A)(3p_A - 1) > 0.$$

This proves monotonicity for increasing $N_{11}$. Hence, the monotonicity property is met.

**NMI and VI.** Let $B'$ be obtained by a perfect split of a cluster $B_1$ into $B'_1, B'_2$. Note that this increases the entropy of the candidate while keeping the joint entropy constant. Let us denote this increase in the candidate entropy by the conditional entropy $H(B'|B) = H(B') - H(B) > 0$. Now, for NMI, the numerator increases by $H(B'|B)$ while the denominator increases by at most $H(B'|B)$ (dependent on $H(A)$ and the specific normalization that is used). Therefore, NMI increases. Similarly, VI decreases by $H(B'|B)$. Concluding, both NMI and VI are monotonous w.r.t. perfect splits. Now let $B''$ be obtained by a perfect merge of $B_1, B_2$ into $B''_1$. This results in a difference of the entropy of the candidate $H(B'') - H(B) = -H(B|B'') < 0$. The joint entropy decreases by the same amount, so that the mutual information remains unchanged. Therefore, the numerator of NMI remains unchanged while the denominator may or may not change, depending on the normalization. For min- or max-normalization, it may remain unchanged while for any other average it increases. Hence, NMI does not satisfy monotonicity w.r.t. perfect merges for min- and max-normalization but does satisfy this for average-normalization. For VI, the distance will decrease by $H(B|B'')$ so that it indeed satisfies monotonicity w.r.t. perfect merges.

**AMI.** Let $B'$ be obtained by splitting a cluster $B_1$ into $B'_1, B'_2$. This split increases the mutual information by $H(B'|B) - H(A \otimes B'|A \otimes B)$. Recall the definition of the meet $A \otimes B$ from C.4 and note that the joint entropy equals $H(A \otimes B)$. For a perfect split we have $H(A \otimes B'|A \otimes B) = 0$. The expected mutual information changes with

$$\mathbf{E}_{A' \sim \mathcal{C}(S(A))}[M(A', B') - M(A', B)] = H(B'|B) - \mathbf{E}_{A' \sim \mathcal{C}(S(A))}[H(A' \otimes B') - H(A' \otimes B)],$$

where we choose to randomize $A$ instead of $B'$ and $B$ for simplicity. Note that for all $A'$,

$$H(A' \otimes B) - H(A' \otimes B') = H(A' \otimes B'|A' \otimes B) \geq 0,$$

with equality if and only if the split is a perfect split w.r.t. $A'$. Unless $A$ consists exclusively of singleton clusters, there is a positive probability that this split is not perfect, so that the expected value is positive. Furthermore, for the normalization term, we have $\sqrt{H(A)H(B')} < \sqrt{H(A)H(B)} + H(B'|B)$. Combining this, we get

$$\mathrm{AMI}(A, B')$$
$$= \frac{M(A, B) - \mathbf{E}_{A' \sim \mathcal{C}(S(A))}[M(A', B)] + \mathbf{E}_{A' \sim \mathcal{C}(S(A))}[H(A' \otimes B'|A' \otimes B)]}{\sqrt{H(A)H(B')} - H(B'|B) - \mathbf{E}_{A' \sim \mathcal{C}(S(A))}[M(A', B)] + \mathbf{E}_{A' \sim \mathcal{C}(S(A))}[H(A' \otimes B'|A' \otimes B)]}$$
$$> \frac{M(A, B) - \mathbf{E}_{A' \sim \mathcal{C}(S(A))}[M(A', B)] + \mathbf{E}_{A' \sim \mathcal{C}(S(A))}[H(A' \otimes B'|A' \otimes B)]}{\sqrt{H(A)H(B)} - \mathbf{E}_{A' \sim \mathcal{C}(S(A))}[M(A', B)] + \mathbf{E}_{A' \sim \mathcal{C}(S(A))}[H(A' \otimes B'|A' \otimes B)]}$$
$$> \frac{M(A, B) - \mathbf{E}_{A' \sim \mathcal{C}(S(A))}[M(A', B)]}{\sqrt{H(A)H(B)} - \mathbf{E}_{A' \sim \mathcal{C}(S(A))}[M(A', B)]} = \mathrm{AMI}(A, B).$$

This proves that AMI satisfies monotonicity w.r.t. perfect splits.

Now let $B''$ be obtained by a perfect merge of $B_1, B_2$ into $B''_1$. Again, we have $H(B'') - H(B) = -H(B|B'' < 0)$ and $M(A, B'') = M(A, B)$. Let $A' \sim \mathcal{C}(S(A))$ (again, randomizing $A$ instead of $B$ and $B''$ for simplicity), then $H(A' \otimes B'') \geq H(A' \otimes B) - H(B|B'')$ with equality if and only if $B''$ is a perfect merge w.r.t. $A'$ which happens with probability strictly less than 1 (unless $A$ consists of a single cluster). Therefore, as long as $k_A > 1$, the expected mutual information decreases. For the normalization, we have $\sqrt{H(A)H(B'')} < \sqrt{H(A)H(B)}$. Hence,

$$\mathrm{AMI}(A, B'') = \frac{M(A, B'') - \mathbf{E}_{A' \sim \mathcal{C}(S(A))}[M(A', B'')]}{\sqrt{H(A)H(B'')} - \mathbf{E}_{A' \sim \mathcal{C}(S(A))}[M(A', B'')]}$$
$$= \frac{M(A, B) - \mathbf{E}_{A' \sim \mathcal{C}(S(A))}[M(A', B'')]}{\sqrt{H(A)H(B'')} - \mathbf{E}_{A' \sim \mathcal{C}(S(A))}[M(A', B'')]}$$
$$> \frac{M(A, B) - \mathbf{E}_{A' \sim \mathcal{C}(S(A))}[M(A', B)]}{\sqrt{H(A)H(B'')} - \mathbf{E}_{A' \sim \mathcal{C}(S(A))}[M(A', B)]}$$
$$> \frac{M(A, B) - \mathbf{E}_{A' \sim \mathcal{C}(S(A))}[M(A', B)]}{\sqrt{H(A)H(B)} - \mathbf{E}_{A' \sim \mathcal{C}(S(A))}[M(A', B)]}$$
$$= \mathrm{AMI}(A, B).$$

**BCubed.** Note that a perfect merge increases BCubed recall while leaving BCubed precision unchanged and that a perfect split increases precision while leaving recall unchanged. Hence, the harmonic mean increases.

### C.4.2 NEGATIVE CASES

**FMeasure.** We give a numerical counter-example: consider $A = \{\{0, \ldots, 6\}\}, B = \{\{0, 1, 2, 3\}, \{4, 5\}, \{6\}\}$ and merge the last two clusters to obtain $B' = \{\{0, 1, 2, 3\}, \{4, 5, 6\}\}$. Then, the FMeasure remains unchanged and equal to 0.73, violating monotonicity w.r.t. perfect merges.

**FNMI** We will give the following numerical counter-example: Consider $A = \{\{0, 1\}, \{2\}, \{3\}\}, B = \{\{0\}, \{1\}, \{2, 3\}\}$ and merge the first two clusters to obtain $B' = \{\{0, 1\}, \{2, 3\}\}$. This results in

$$\text{FNMI}(A, B) \approx 0.67 > 0.57 \approx \text{FNMI}(A, B').$$

This non-monotonicity is caused by the penalty factor that equals 1 for the pair $A, B$ and equals $\exp(-1/3) \approx 0.72$ for $A, B'$.

**SMI.** For this numerical counter-example we rely on the Matlab-implementation of the index by its original authors (Romano et al., 2014). Let $A = \{\{0, \ldots, 4\}, \{5\}\}, B = \{\{0, 1\}, \{2, 3\}, \{4\}, \{5\}\}$ and consider merging the two clusters resulting in $B' = \{\{0, 1, 2, 3\}, \{4\}, \{5\}\}$. The index remains unchanged and equals 2 before and after the merge.

**Wallace.** Let $k_A = 1$ and let $k_B > 1$. Then any merge of $B$ is a perfect merge, but no increase occurs since $W_1(A, B) = 1$.

### C.5 CONSTANT BASELINE

### C.5.1 POSITIVE CASES

**AMI and SMI.** Both of these indices satisfy the constant baseline by construction since the expected mutual information is subtracted from the actual mutual information in the numerator.

**Adjusted Rand, Correlation Coefficient and Sokal&Sneath.** These indices all satisfy ACB while being $P_{AB}$-linear for fixed $p_A, p_B$. Therefore, the expected value equals the asymptotic constant.

### C.5.2 NEGATIVE CASES

For all the following indices, we will analyse the following counter-example. Let $|V| = n, k_A = k_B = n - 1$. For each index, we will compute the expected value and show that it is not constant. All of these indices satisfy the maximal agreement property and maximal agreement is achieved with probability $1/N$ (the probability that the single intra-pair of $A$ coincides with the single intra-pair of $B$). Furthermore, each case where the intra-pairs do not coincide will result in the same contingency variables and hence the same value of the index. We will refer to this value as $c_n(I)$. Therefore, the expected value will only have to be taken over two values and will be given by

$$\mathbf{E}[I(A, B)] = \frac{1}{N} c_{\max} + \frac{N - 1}{N} c_n(I).$$

For each of these indices we will conclude that this is a non-constant function of $n$ so that the index does not satisfy the constant baseline property.

**Jaccard and Dice.** For both these indices we have $c_{\max} = 1$ and $c_n(I) = 0$ (as $N_{11} = 0$ whenever the intra-pairs do not coincide). Hence, $\mathbf{E}[I(A, B)] = \frac{1}{N}$, which is not constant.

**Rand and Wallace.** As both functions are linear in $N_{11}$ for fixed $m_A = N_{11} + N_{10}, m_B = N_{11} + N_{01}$, we can compute the expected value by simply substituting $N_{11} = m_A m_B / N$. This will result in expected values $1 - 2/N + 2/N^2$ and $1/N$ for Rand and Wallace respectively, which are both non-constant.

**Correlation distance.** Here $c_{\max} = 0$ and

$$c_n(CD) = \frac{1}{\pi} \arccos\left(\frac{0 - 1/N^2}{(N-1)/N^2}\right),$$

so that the expected value will be given by

$$\mathbf{E}[CD(A,B)] = \frac{N-1}{N\pi} \arccos\left(-\frac{1}{N-1}\right).$$

This is non-constant (it evaluates to $0.44, 0.47$ for $n = 3, 4$ respectively). Note that this expected value converges to $\frac{1}{2}$ for $n \to \infty$, which is indeed the asymptotic baseline of the index.

**FNMI and NMI.** Note that in this case $k_A = k_B$ so that the penalty term of FNMI will equal $1$ and FNMI will coincide with NMI. Again $c_{\max} = 1$. For the case where the intra-pairs do not coincide, the joint entropy will equal $H(A,B) = \ln(n)$ while each of the marginal entropies will equal

$$H(A) = H(B) = \frac{n-2}{n} \ln(n) + \frac{2}{n} \ln(n/2) = \ln(n) - \frac{2}{n} \ln(2).$$

This results in

$$c_n(NMI) = \frac{2H(A) - H(A,B)}{H(A)} = 1 - \frac{2\ln(n)}{n\ln(n) - 2\ln(2)},$$

and the expected value will be given by the non-constant

$$\mathbf{E}[NMI(A,B)] = 1 - \frac{N-1}{N} \frac{2\ln(n)}{n\ln(n) - 2\ln(2)}.$$

Note that as $H(A) = H(B)$, all normalizations of MI will be equal so that this counter-example proves that none of the variants of (F)NMI satisfy the constant baseline property.

**Variation of Information.** In this case $c_{\max} = 0$. We will use the entropies from the NMI-computations to conclude that

$$\mathbf{E}[VI(A,B)] = \frac{N-1}{N}(2H(A,B) - H(A) - H(B)) = \frac{N-1}{N} \frac{4}{n} \ln(2),$$

which is again non-constant.

**F-measure.** Here $c_{\max} = 1$. In the case where the intra-pairs do not coincide, all contingency variables will be either one or zero so that both recall and precision will equal $1 - 1/n$ so that $c_n(FM) = 1 - 1/n$. This results in the following non-constant expected value

$$\mathbf{E}[FM(A,B)] = 1 - \frac{N-1}{N} \frac{1}{n}.$$

Note that because recall equals precision in both cases, this counter-example also works for other averages than the harmonic average.

**BCubed.** Again $c_{\max} = 1$. In the other case, the recall and precision will again be equal. Because for BCubed, the contribution of cluster $i$ is given by $\frac{1}{n} \max\{n_{ij}^2\}/|A_i|$, the contributions of the one- and two-clusters will be given by $\frac{1}{n}, \frac{1}{2n}$ respectively. Hence, $c_n(BC) = \frac{n-2}{n} + \frac{1}{2n} = 1 - \frac{3}{2n}$ and we get the non-constant

$$\mathbf{E}[BC(A,B)] = 1 - \frac{N-1}{N} \cdot \frac{3}{2n}.$$

We note that again, this counter-example can be extended to non-harmonic averages of the BCubed recall and precision.

# D FUTHER ANALYSIS OF CONSTANT BASELINE PROPERTY

## D.1 ANALYSIS OF EXACT CONSTANT BASELINE PROPERTY

Let us show that the definition of the constant baseline applies not only to uniform (within a given sizes specification) distribution but also to all symmetric distributions over clusterings.

**Definition 11.** *We say that a distribution over clusterings $\mathcal{B}$ is* element-symmetric *if for every two clusterings $B$ and $B'$ that have the same cluster-sizes, $\mathcal{B}$ returns $B$ and $B'$ with equal probabilities.*

**Lemma 3.** *Let $I$ be an index with a constant baseline as defined in Definition 7, let $A$ be a clustering with $1 < k_A < n$ and let $\mathcal{B}$ be an element-symmetric distribution. Then $\mathbf{E}_{B \sim \mathcal{B}}[I(A, B)] = c_{base}$.*

*Proof.* We write

$$\mathbf{E}_{B \sim \mathcal{B}}[I(A, B)] = \sum_s \mathbf{P}_{B \sim \mathcal{B}}(S(B) = s) \, \mathbf{E}_{B \sim \mathcal{B}}[I(A, B)|S(B) = s]$$

$$= \sum_s \mathbf{P}_{B \sim \mathcal{B}}(S(B) = s) \, \mathbf{E}_{B \sim \mathcal{C}(s)}[I(A, B)]$$

$$= \sum_s \mathbf{P}_{B \sim \mathcal{B}}(S(B) = s) \, c_{\text{base}} = c_{\text{base}},$$

where the sum ranges over cluster-sizes of $n$ elements. $\qquad\square$

## D.2 ANALYSIS OF ASYMPTOTIC CONSTANT BASELINE PROPERTY

**Definition 12.** *An index $I$ is said to be* scale-invariant*, if it can be expressed as a continuous function of the three variables $p_A := m_A/N, p_B := m_B/N$ and $p_{AB} := N_{11}/N$.*

All indices in Table 3 are scale-invariant. For such indices, we will write $I^{(p)}(p_{AB}, p_A, p_B)$. Note that when $B \sim \mathcal{C}(s)$ for some $s$, the values $p_A, p_B$ are constants while $p_{AB}$ is a random variable. Therefore, we further write $P_{AB}$ to stress that this is a random variable.

**Theorem 2.** *Let $I$ be a scale-invariant pair-counting index, and consider a sequence of clusterings $A^{(n)}$ and cluster-size specifications $s^{(n)}$. Let $N_{11}^{(n)}, N_{10}^{(n)}, N_{01}^{(n)}, N_{00}^{(n)}$ be the corresponding pair-counts. Then, for any $\varepsilon > 0$, as $n \to \infty$,*

$$\mathbf{P}\left(\left|I\left(N_{11}^{(n)}, N_{10}^{(n)}, N_{01}^{(n)}, N_{00}^{(n)}\right) - I\left(\overline{N_{11}^{(n)}}, \overline{N_{10}^{(n)}}, \overline{N_{01}^{(n)}}, \overline{N_{00}^{(n)}}\right)\right| > \varepsilon\right) \to 0.$$

*Proof.* We prove the equivalent statement

$$I^{(p)}\left(P_{AB}^{(n)}, p_A^{(n)}, p_B^{(n)}\right) - I^{(p)}\left(p_A^{(n)} p_B^{(n)}, p_A^{(n)}, p_B^{(n)}\right) \xrightarrow{P} 0.$$

We first prove that $P_{AB}^{(n)} - p_A^{(n)} p_B^{(n)} \xrightarrow{P} 0$ so that the above follows from the continuous mapping theorem. Chebychev's inequality gives

$$\mathbf{P}\left(|P_{AB}^{(n)} - p_A^{(n)} p_B^{(n)}| > \varepsilon\right) \leq \frac{1}{\binom{n}{2}^2 \varepsilon^2} \text{Var}\left(N_{11}^{(n)}\right) \to 0.$$

The last step follows from the fact that $\text{Var}(N_{11}) = o(n^4)$, as we will prove in the remainder of this section. Even though in the definition, $A$ is fixed while $B$ is randomly permuted, it is convenient to equivalently consider both clusterings are randomly permuted for this proof.

We will show that $\text{Var}(N_{11}) = o(n^4)$. To compute the variance, we first inspect the second moment. Let $A(S)$ denote the indicator function of the event that all elements of $S \subset V$ are in the same cluster in $A$. Define $B(S)$ similarly and let $AB(S) = A(S)B(S)$. Let $e, e_1, e_2$ range over subsets of $V$ of

size 2. We write

$$
N_{11}^2 = \left( \sum_e AB(e) \right)^2
$$

$$
= \sum_{e_1, e_2} AB(e_1) AB(e_2)
$$

$$
= \sum_{|e_1 \cap e_2| = 2} AB(e_1) AB(e_2) + \sum_{|e_1 \cap e_2| = 1} AB(e_1) AB(e_2) + \sum_{|e_1 \cap e_2| = 0} AB(e_1) AB(e_2)
$$

$$
= N_{11} + \sum_{|e_1 \cap e_2| = 1} AB(e_1 \cup e_2) + \sum_{e_1 \cap e_2 = \emptyset} AB(e_1) AB(e_2).
$$

We take the expectation

$$
\mathbf{E}[N_{11}^2] = \mathbf{E}[N_{11}] + 6 \binom{n}{3} \mathbf{E}[AB(\{v_1, v_2, v_3\})] + \binom{n}{2} \binom{n-2}{2} \mathbf{E}[AB(e_1) AB(e_2)],
$$

where $v_1, v_2, v_3 \in V$ distinct and $e_1 \cap e_2 = \emptyset$. The first two terms are obviously $o(n^4)$. We inspect the last term

$$
\binom{n}{2} \binom{n-2}{2} \mathbf{E}[AB(e_1) AB(e_2)]
$$

$$
= \binom{n}{2} \sum_{i,j} \mathbf{P}(e_1 \subset A_i \cap B_j) \times \binom{n-2}{2} \mathbf{E}[AB(e_2) | e_1 \subset A_i \cap B_j]. \tag{3}
$$

Now we rewrite $\mathbf{E}[N_{11}]^2$ to

$$
\mathbf{E}[N_{11}]^2 = \binom{n}{2} \sum_{i,j} \mathbf{P}(e_1 \subset A_i \cap B_j) \binom{n}{2} \mathbf{E}[AB(e_2)].
$$

Note that $\binom{n}{2} \mathbf{E}[AB(e_2)] > \binom{n-2}{2} \mathbf{E}[AB(e_2)]$ so that the difference between (3) and $\mathbf{E}[N_{11}]^2$ can be bounded by

$$
\binom{n}{2} \binom{n-2}{2} \sum_{i,j} \mathbf{P}(e_1 \subset A_i \cap B_j) \cdot (\mathbf{E}[AB(e_2) | e_1 \subset A_i \cap B_j] - \mathbf{E}[AB(e_2)]).
$$

As $\binom{n}{2} \binom{n-2}{2} = O(n^4)$, what remains to be proven is

$$
\sum_{i,j} \mathbf{P}(e_1 \subset A_i \cap B_j) \cdot (\mathbf{E}[AB(e_2) | e_1 \subset A_i \cap B_j] - \mathbf{E}[AB(e_2)]) = o(1).
$$

Note that it is sufficient to prove that

$$
\mathbf{E}[AB(e_2) | e_1 \subset A_i \cap B_j] - \mathbf{E}[AB(e_2)] = o(1),
$$

for all $i, j$. Note that $\mathbf{E}[AB(e_2)] = m_A m_B / N^2$, while

$$
\mathbf{E}[AB(e_2) | e_1 \subset A_i \cap B_j] = \frac{(m_A - (2a_i - 3))(m_B - (2b_j - 3))}{(N - (2n - 3))^2}.
$$

Hence, the difference will be given by

$$
\frac{(m_A - (2a_i - 3))(m_B - (2b_j - 3))}{(N - (2n - 3))^2} - \frac{m_A m_B}{N^2}
$$

$$
= \frac{N^2 (m_A - (2a_i - 3))(m_B - (2b_j - 3))}{N^2 (N - (2n - 3))^2} - \frac{(N - (2n - 3))^2 m_A m_B}{N^2 (N - (2n - 3))^2}
$$

$$
= \frac{N^2 ((2a_i - 3)(2b_j - 3) - m_A(2b_j - 3) - m_B(2a_i - 3))}{N^2 (N - (2n - 3))^2} + \frac{m_A m_B (2N(2n - 3) - (2n - 3)^2)}{N^2 (N - (2n - 3))^2}
$$

$$
= \frac{((2a_i - 3)(2b_j - 3) - m_A(2b_j - 3) - m_B(2a_i - 3))}{(N - (2n - 3))^2} + \frac{m_A m_B}{N^2} \frac{(2N(2n - 3) - (2n - 3)^2)}{(N - (2n - 3))^2}
$$

$$
= \frac{O(n^3)}{(N - (2n - 3))^2} + \frac{m_A m_B}{N^2} \frac{O(n^3)}{N^2 (N - (2n - 3))^2}
$$

$$
= o(1),
$$

as required.

$\square$

## D.3 STATISTICAL TESTS FOR CONSTANT BASELINE

In this section, we provide two statistical tests: one test to check whether an index $I$ satisfies the constant baseline property and another to check whether $I$ has a selection bias towards certain cluster sizes.

**Checking constant baseline.** Given a reference clustering $A$ and a number of cluster sizes specifications $s_1, \ldots, s_k$, we test the null hypothesis that

$$\mathbf{E}_{B \sim \mathcal{C}(s_i)}[I(A, B)]$$

is constant in $i = 1, \ldots, k$. We do so by using one-way Analysis Of Variance (ANOVA). For each cluster sizes specification, we generate $r$ clusterings. Although ANOVA assumes the data to be normally distributed, it is known to be robust for sufficiently large groups (i.e., large $r$).

**Checking selection bias.** In (Romano et al., 2014) it is observed that some indices with a constant baseline do have a *selection bias*; when we have a pool of random clusterings of various sizes and select the one that has the highest score w.r.t. a reference clustering, there is a bias of selecting certain cluster sizes. We test this bias in the following way: given a reference clustering $A$ and cluster sizes specifications $s_1, \ldots, s_k$, we repeatedly generate $B_1 \sim \mathcal{C}(s_1), \ldots, B_k \sim \mathcal{C}(s_k)$. The null-hypothesis will be that each of these clusterings $B_i$ has an equal chance of maximizing $I(A, B_i)$. We test this hypothesis by generating $r$ pools and using the Chi-squared test.

We emphasize that these statistical tests cannot prove whether an index satisfies the property or has a bias. Both will return a confidence level $p$ with which the null hypothesis can be rejected. Furthermore, for an index to not have these biases, the null hypothesis should be true for all choices of $A, s_1, \ldots, s_k$, which is impossible to verify statistically.

The statistical tests have been implemented in Python and the code supplements the submission. We applied the tests to the indices of Tables 2 and 3. We chose $n = 50, 100, 150, \ldots, 1000$ and $r = 500$. For the cluster sizes, we define the *balanced cluster sizes $BS(n, k)$* to be the cluster-size specification for $k$ clusters of which $n - k * \lfloor n/k \rfloor$ clusters have size $\lceil n/k \rceil$ while the remainder have size $\lfloor n/k \rfloor$. Then we choose $A^{(n)}$ to be a clustering with sizes $BS(n, \lfloor n^{0.5} \rfloor)$ and consider candidates with sizes $s_1^{(n)} = BS(n, \lfloor n^{0.25} \rfloor), s_2^{(n)} = BS(n, \lfloor n^{0.5} \rfloor), s_3^{(n)} = BS(n, \lfloor n^{0.75} \rfloor)$. For each $n$, the statistical test returns a $p$-value. We use Fisher's method to combine these $p$-values into one single $p$-value and then reject the constant baseline if $p < 0.05$. The obtained results agree with Tables 2 and 3 except for Correlation Distance, which is so close to having a constant baseline that the tests are unable to detect it.

## D.4 ILLUSTRATING SIGNIFICANCE OF CONSTANT BASELINE

In this section, we conduct two experiments illustrating the biases of various indices. We perform two experiments that allow us to identify the direction of the bias in different situations. Our reference clustering corresponds to the expert-annotated clustering of the production experiment described in Section 3 and Appendix F.3, where $n = 924$ items are grouped into $k_A = 431$ clusters (305 of them consist of a single element).

In the first experiment, we randomly cluster the items into $k$ approximately equally sized clusters for various $k$. Figure 3 shows the averages and $90\%$ confidence bands for each index. It can be seen that some indices (e.g., NMI and Rand) have a clear increasing baseline while others (e.g., Jaccard and VI) have a decreasing baseline. In contrast, all unbiased indices have a constant baseline.

In Section 4.6 we argued that these biases could not be described in terms of the number of clusters alone. Our second experiment illustrates that the bias also heavily depends on the sizes of the clusters. In this case, items are randomly clustered into 32 clusters, 31 of which are "small" clusters of size $s$ while one cluster has size $n - 31 \cdot s$, where $s$ is varied between 1 and 28. We see that the biases are clearly visible. This shows that, even when fixing the number of clusters, biased indices may heavily distort an experiment's outcome.

Finally, recall that we have proven that the baseline of CD is only asymptotically constant. Figures 3 and 4 show that for practical purposes its baseline can be considered constant.

## E  ADDITIONAL RESULTS

### E.1  PROOF OF THEOREM 1

Let $B'$ be an $A$-consistent improvement of $B$. We define

$$B \otimes B' = \{B_j \cap B'_{j'} | B_j \in B, B'_{j'} \in B', B_j \cap B'_{j'} \neq \emptyset\}$$

and show that $B \otimes B'$ can be obtained from $B$ by a sequence of perfect splits, while $B'$ can be obtained from $B \otimes B'$ by a sequence of perfect merges. Indeed, the assumption that $B'$ does not introduce new disagreeing pairs guarantees that any $B_j \in B$ can be split into $B_j \cap B'_1, \ldots, B_j \cap B'_{k_{B'}}$ without splitting over any intra-cluster pairs of $A$. Let us prove that $B'$ can be obtained from $B \otimes B'$ by perfect merges. Suppose there are two $B''_1, B''_2 \in B \otimes B'$ such that both are subsets of some $B'_{j'}$. Assume that this merge is not perfect, then there must be $v \in B''_1, w \in B''_2$ such that $v, w$ are in different clusters of $A$. As $v, w$ are in the same cluster of $B'$, it follows from the definition of $B \otimes B'$ that $v, w$ must be in different clusters of $B$. Hence, $v, w$ is an inter-cluster pair in both $A$ and $B$, while it is an intra-cluster pair of $B'$, contradicting the assumption that $B'$ is an $A$-consistent improvement of $B$. This concludes the proof.

### E.2  CORRELATION DISTANCE IS A DISTANCE

**Theorem.** *The Correlation Distance is indeed a distance.*

*Proof.* A proof of this is given in Van Dongen & Enright (2012). We give an alternative proof that allows for a geometric interpretation. First we map each partition $A$ to an $N$-dimensional vector on the unit sphere by

$$\vec{u}(A) := \begin{cases} \frac{1}{\sqrt{N}}\mathbf{1} & \text{if } k_A = 1, \\ \frac{\vec{A}-p_A\mathbf{1}}{\|\vec{A}-p_A\mathbf{1}\|} & \text{if } 1 < k_A < n, \\ -\frac{1}{\sqrt{N}}\mathbf{1} & \text{if } k_A = n, \end{cases}$$

where $\mathbf{1}$ is the $N$-dimensional all-one vector and $\vec{A}$ is the binary vector representation of a partition introduced in Section 2. Straightforward computation gives $\|\vec{A} - p_A\mathbf{1}\| = \sqrt{Np_A(1-p_A)}$, and standard inner product

$$\langle \vec{A} - p_A\mathbf{1}, \vec{B} - p_B\mathbf{1} \rangle = N(p_{AB} - p_A p_B),$$

so that indeed

$$\frac{\langle \vec{A} - p_A\mathbf{1}, \vec{B} - p_B\mathbf{1} \rangle}{\|\vec{A} - p_A\mathbf{1}\|\|\vec{B} - p_B\mathbf{1}\|} = CC^{(p)}(p_{AB}, p_A, p_B).$$

It is a well-known fact that the inner product of two vectors of unit length corresponds to the cosine of their angle. Hence, taking the arccosine gives us the angle. The angle between unit vectors corresponds to the distance along the unit sphere. As $\vec{u}$ is an injection from the set of partitions to points on the unit sphere, we may conclude that this index is indeed a distance on the set of partitions. □

### E.3  DEVIATION OF CD FROM CONSTANT BASELINE

**Theorem.** *Given ground truth $A$ with a number of clusters $1 < k_A < n$, a cluster-size specification $s$ and a random partition $B \sim \mathcal{C}(s)$, the expected difference between Correlation Distance and its baseline is given by*

$$\mathbf{E}_{B \sim \mathcal{C}(s)}[\text{CD}(A, B)] - \frac{1}{2} = -\frac{1}{\pi}\sum_{k=1}^{\infty}\frac{(2k)!}{2^{2k}(k!)^2}\frac{\mathbf{E}_{B \sim \mathcal{C}(s)}[\text{CC}(A, B)^{2k+1}]}{2k+1}.$$

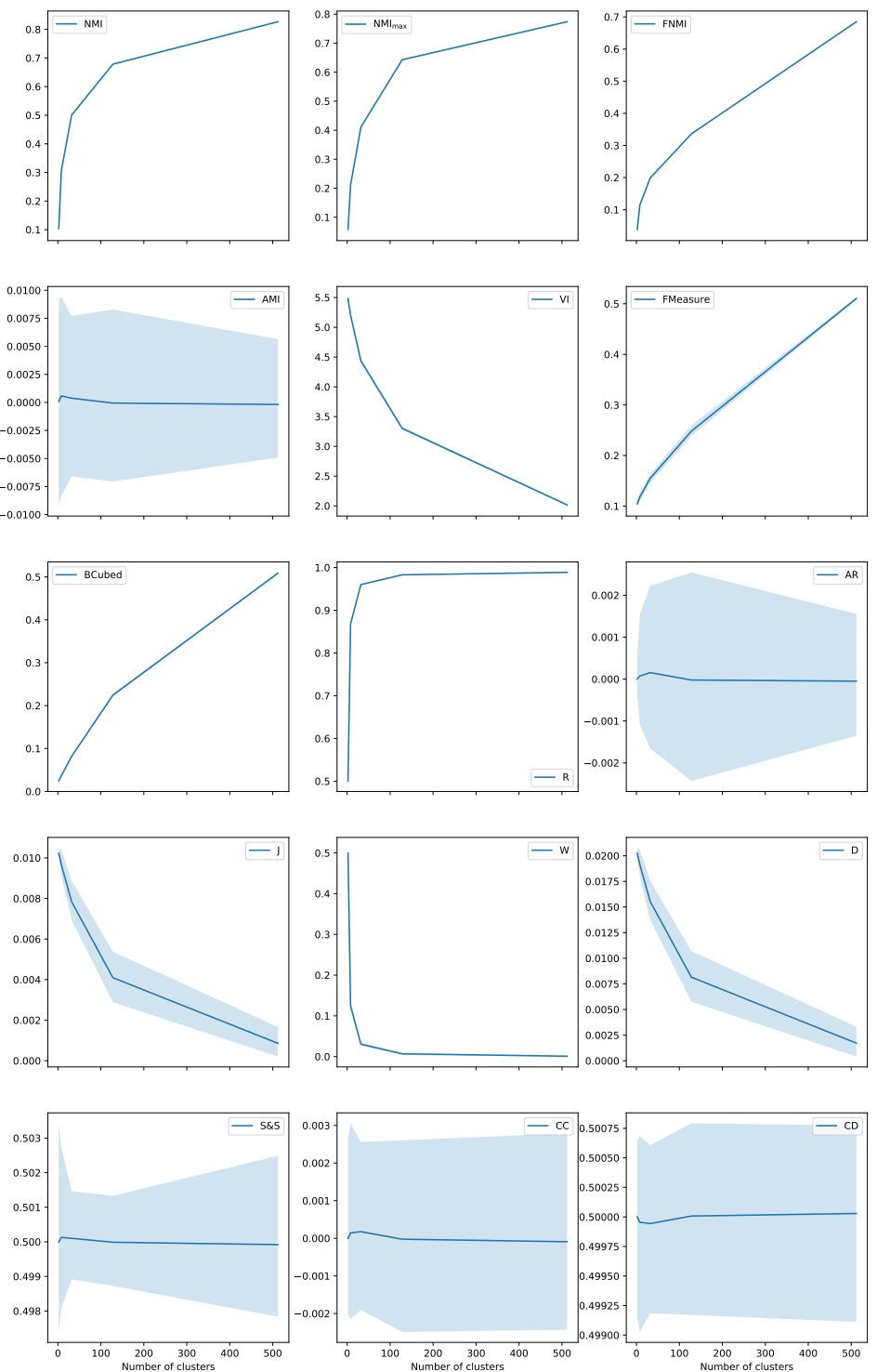

Figure 3: The reference clustering of Appendix F.3 ($n = 924$ and $k_A = 431$) is compared to random clusterings. Each clustering consists of $k$ approximately equally-sized clusters, where $k$ is varied between $2$ and $512$. For each $k$, 200 random clusterings are generated. For each index, we plot the average score, along with a $90\%$ confidence band.

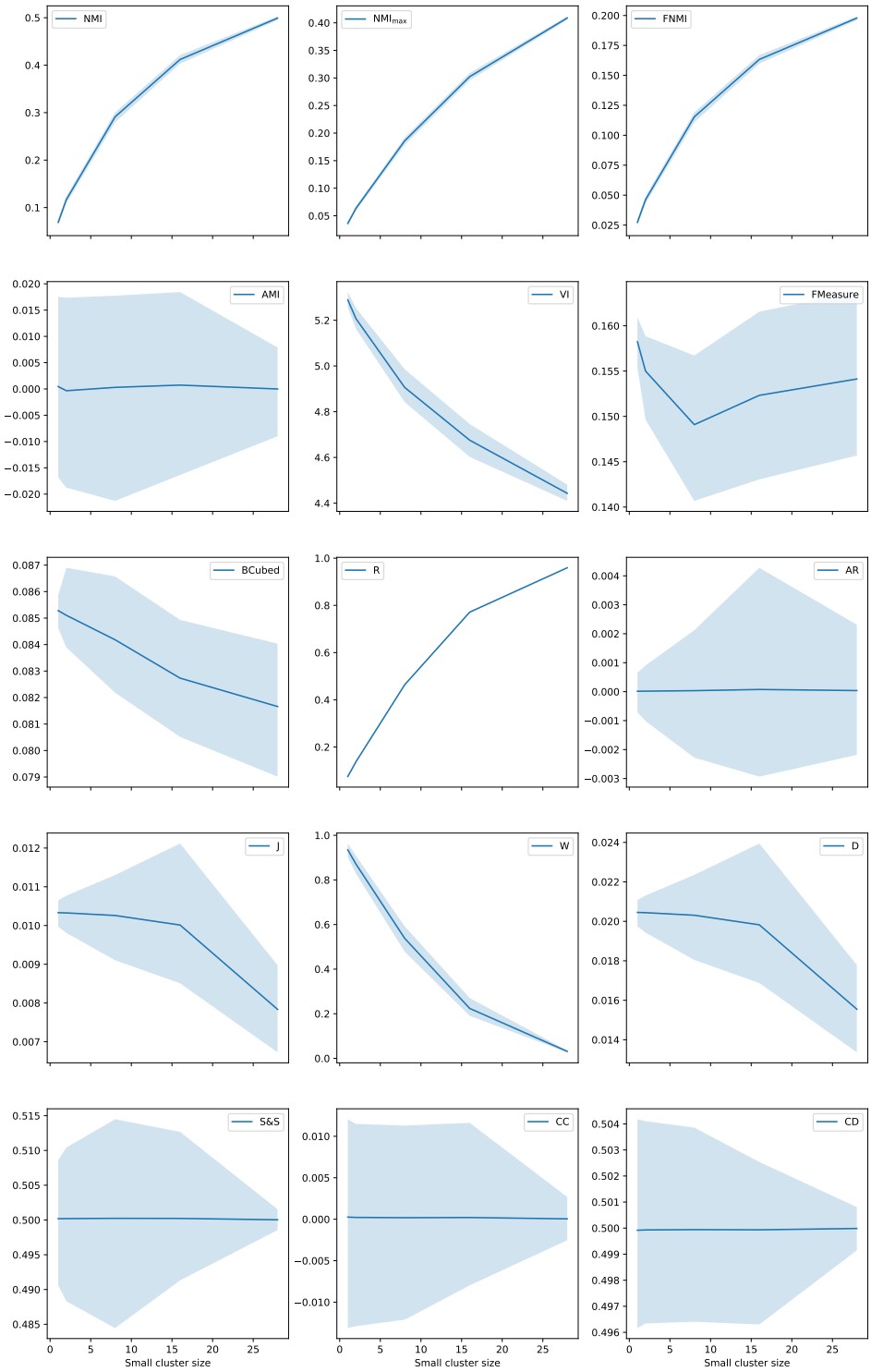

Figure 4: The reference clustering of Appendix F.3 ($n = 924$ and $k_A = 431$) is compared to random clusterings. Each clustering consists of 31 "small" clusters of size $s$ while the last cluster has size $924 - 31 \cdot s$, where $s$ is varied between 1 and 28. For each $s$, 200 random clusterings are generated. For each index, we plot the average score, along with a $90\%$ confidence band.

*Proof.* We take the Taylor expansion of the arccosine around $CC(A, B) = 0$ and get

$$\mathrm{CD}(A, B) = \frac{1}{2} - \frac{1}{\pi} \sum_{k=0}^{\infty} \frac{(2k)!}{2^{2k}(k!)^2} \frac{CC(A, B)^{2k+1}}{2k+1}.$$

We take the expectation of both sides and note that the first moment of CC equals zero, so the starting index is $k = 1$. □

For $B \sim \mathcal{C}(s)$ and large $n$, the value $CC(A, B)$ will be concentrated around 0. This explains that in practice, the mean tends to be very close to the asymptotic baseline.

### E.4 COMPARISON WITH LEI ET AL. (2017)

Lei et al. (2017) describe the following biases for cluster similarity indices: *NCinc* — the average value for a random guess increases monotonically with the Number of Clusters (NC) of the candidate; *NCdec* — the average value for a random guess decreases monotonically with the number of clusters, and *GTbias* — the direction of the monotonicity depends on the specific Ground Truth (GT), i.e., on the reference partition. In particular, the authors conclude from numerical experiments that Jaccard suffers from NCdec and analytically prove that Rand suffers from GTbias, where the direction of the bias depends on the quadratic entropy of the ground truth clustering. Here we argue that these biases are not well defined, suggest replacing them by well-defined analogs, and show how our analysis allows to easily test indices on these biases.

We argue that the quantity of interest should not be the *number of clusters*, but the *number of intra-cluster pairs* of the candidate. Theorem 2 shows that the asymptotic value of the index depends on the number of intra-cluster pairs of both clusterings. The key insight is that more clusters do not necessarily imply fewer intra-cluster pairs. For example, let $s$ denote a cluster-sizes specification for 3 clusters each of size $\ell > 2$. Now let $s'$ be the cluster-sizes specification for one cluster of size $2\ell$ and $\ell$ clusters of size 1. Then, any $B \sim \mathcal{C}(s)$ will have 3 clusters and $3\binom{\ell}{2}$ intra-cluster pairs while any $B' \sim \mathcal{C}(s')$ will have $\ell + 1 > 3$ clusters and $\binom{2\ell}{2} > 3\binom{\ell}{2}$ intra-cluster pairs. For any ground truth $A$ with cluster-sizes $s$, we have $\mathbf{E}[J(A, B)] < \mathbf{E}[J(A, B')]$ because of a larger amount of intra-cluster pairs In contrast, Lei et al. (2017) classifies Jaccard as an NCdec index, so that the expected value should increase, contradicting the definition of NCdec. The NPinc and NPdec biases that are defined in Definition 9 are sound versions of these NCinc and NCdec biases because they depend on the expected number of agreeing pairs. This allows to analytically determine which bias a given pair-counting index has.

## F EXPERIMENT

### F.1 SYNTHETIC EXPERIMENT

In this experiment, we construct several simple examples to illustrate the inconsistency among the indices. Recall that two indices $I_1$ and $I_2$ are inconsistent for a triplet of partitions $(A, B_1, B_2)$ if $I_1(A, B_1) > I_1(A, B_2)$ but $I_2(A, B_1) < I_2(A, B_2)$.

We take all indices from Tables 2 and 3 and construct several triplets of partitions to distinguish them all. Let us note that the pairs Dice vs Jaccard and CC vs CD cannot be inconsistent since they are monotonically transformable to each other. Also, we do not compare with SMI since it is much more computationally complex than all other indices. Thus, we end up with 13 indices and are looking for simple inconsistency examples.

The theoretical minimum of examples needed to find inconsistency for all pairs of 13 indices is 4. We were able to find such four examples, see Figure 5. In this figure, we show four inconsistency triplets. For each triplet, the shapes (triangle, square, etc.) denote the reference partition $A$. Left and right figures show candidate partitions $B_1$ and $B_2$. In the caption, we specify which similarity indices favor this candidate partition over the other one.

It is easy to see that for each pair of indices, there is a simple example where they disagree. For example, NMI and NMI$_{\mathrm{max}}$ are inconsistent for triplets 3. Also, we know that Jaccard in general

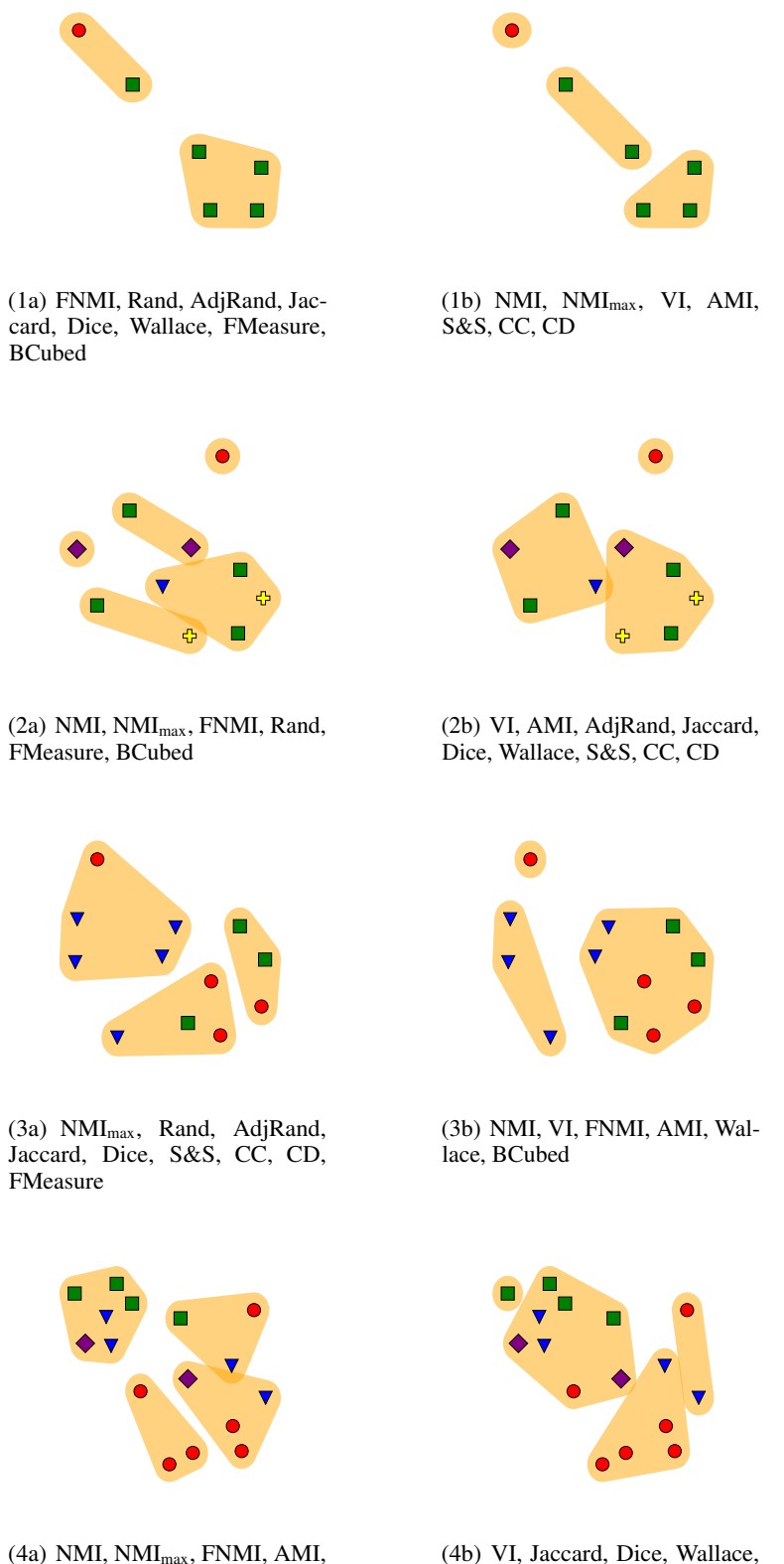

(1a) FNMI, Rand, AdjRand, Jaccard, Dice, Wallace, FMeasure, BCubed

(1b) NMI, $\text{NMI}_{max}$, VI, AMI, S&S, CC, CD

(2a) NMI, $\text{NMI}_{max}$, FNMI, Rand, FMeasure, BCubed

(2b) VI, AMI, AdjRand, Jaccard, Dice, Wallace, S&S, CC, CD

(3a) $\text{NMI}_{max}$, Rand, AdjRand, Jaccard, Dice, S&S, CC, CD, FMeasure

(3b) NMI, VI, FNMI, AMI, Wallace, BCubed

(4a) NMI, $\text{NMI}_{max}$, FNMI, AMI, Rand, AdjRand, CC, CD

(4b) VI, Jaccard, Dice, Wallace, S&S, FMeasure, BCubed

Figure 5: Inconsistency of indices: each row corresponds to a triplet of partitions, shapes denote the reference partitions, the captions indicate which indices favor the corresponding candidate.

Table 6: Inconsistency of indices on real-world clustering datasets, %

| | NMI | NMI$_{max}$ | VI | FNMI | AMI | R | AR | J | W | S&S | CC | FMeas | BCub |
|---|---|---|---|---|---|---|---|---|---|---|---|---|---|
| NMI | – | 5.4 | 40.3 | 17.3 | 9.2 | 13.4 | 15.7 | 35.2 | 68.4 | 20.1 | 18.5 | 31.7 | 32.0 |
| NMI$_{max}$ | | – | 41.1 | 16.5 | 13.2 | 12.5 | 14.1 | 34.3 | 68.8 | 21.1 | 18.9 | 30.3 | 32.4 |
| VI | | | – | 34.7 | 41.8 | 45.2 | 37.6 | 17.1 | 28.8 | 36.0 | 37.2 | 18.1 | 13.6 |
| FNMI | | | | – | 23.3 | 24.0 | 19.0 | 29.9 | 57.0 | 26.7 | 23.8 | 27.5 | 26.7 |
| AMI | | | | | – | 21.1 | 17.3 | 33.3 | 61.3 | 15.1 | 13.6 | 35.0 | 34.4 |
| R | | | | | | – | 15.5 | 35.6 | 71.5 | 21.1 | 20.7 | 32.5 | 35.8 |
| AR | | | | | | | – | 23.5 | 59.4 | 11.7 | 8.3 | 25.3 | 28.1 |
| J | | | | | | | | – | 35.9 | 23.1 | 23.8 | 10.7 | 9.7 |
| W | | | | | | | | | – | 53.5 | 54.8 | 40.7 | 37.4 |
| S&S | | | | | | | | | | – | 3.6 | 26.2 | 27.8 |
| CC | | | | | | | | | | | – | 27.0 | 28.8 |
| FMeas | | | | | | | | | | | | – | 7.7 |
| BCub | | | | | | | | | | | | | – |

favors larger clusters, while Rand and NMI often prefer smaller ones. Hence, they often disagree in this way (see the triplets 2 and 4).

## F.2 Experiments on real datasets

In this section, we test whether the inconsistency affects conclusions obtained in experiments on real data.

For that, we used 16 real-world datasets (GitHub, 2020). We took all real-world datasets available there and removed the ones with categorial features or without the explicit target class field defined. The values of the "target class" field were used as a reference partition. We end up with the following real-world datasets: arrhythmia, balance-scale, ecoli, heart-statlog, letter, segment, vehicle, wdbc, wine, wisc, cpu, iono, iris, sonar, thy, zoo.

On these datasets, we ran 8 well-known clustering algorithms (Scikit-learn, 2020): KMeans, AffinityPropagation, MeanShift, AgglomerativeClustering, DBSCAN, OPTICS, Birch, GaussianMixture. For AgglomerativeClustering, we used 4 different linkage types ('ward', 'average', 'complete', 'single'). For GaussianMixture, we used 4 different covariance types ('spherical', 'diag', 'tied', 'full'). For methods requiring the number of clusters as a parameter (KMeans, Birch, AgglomerativeClustering, GaussianMixture), we took up to 4 different values (less than 4 if some of them are equal): 2, ref-clusters, max(2,ref-clusters/2), min(items, 2·ref-clusters), where ref-clusters is the number of clusters in the reference partition and items is the number of elements in the dataset. For MeanShift, we used the option $cluster\_all = True$. All other settings were default or taken from examples in the sklearn manual.

For all datasets, we calculated all the partitions for all methods described above. We removed all partitions having only one cluster or which raised any calculation error. Then, we considered all possible triplets $A, B_1, B_2$, where $A$ is a reference partition and $B_1$ and $B_2$ are candidates obtained with two different algorithms. We have 8688 such triplets in total. For each triplet, we check whether the indices are consistent. The inconsistency frequency is shown in Table 6. Note that Wallace is highly asymmetrical and does not satisfy most of the properties, so it is not surprising that it is in general very inconsistent with others. However, the inconsistency rates are significant even for widely used pairs of indices such as, e.g., Variation of Information vs NMI (40.3%, which is an extremely high disagreement). Interestingly, the best agreeing indices are S&S and CC which satisfy most of our properties. This means that conclusions made with these indices are likely to be similar.

Actually, one can show that all indices are inconsistent using only one dataset. This holds for 11 out of 16 datasets: heart-statlog, iris, segment, thy, arrhythmia, vehicle, zoo, ecoli, balance-scale, letter, wine. We do not present statistics for individual datasets since we found the aggregated Table 6 to be more useful.

Finally, to illustrate the biases of indices, we compare two KMeans algorithms with $k = 2$ and $k = 2 \cdot$ref-clusters. The comparison is performed on 10 datasets (where both algorithms are successfully completed). The results are shown in Table 7. In this table, biases and inconsistency are clearly

---

[5] The code supplements the submission.

Table 7: Algorithms preferred by different indices

|  | NMI | $NMI_{max}$ | VI | FNMI | AMI | R | AR | J | W | S&S1 | CC | FMeas | BCub |
|---|---|---|---|---|---|---|---|---|---|---|---|---|---|
| $k = 2$ | 2 | 1 | 9 | 4 | 2 | 0 | 4 | 6 | 10 | 3 | 3 | 7 | 7 |
| $k = 2 \cdot$ ref | 8 | 9 | 1 | 6 | 8 | 10 | 6 | 4 | 0 | 7 | 7 | 3 | 3 |

Table 8: Similarity of candidate partitions to the reference one. In bold are the inconsistently ranked pairs of partitions. Some indices are taken with "-" sign, so larger values correspond to better agreement.

|  | $A_{prod}$ | $A_1$ | $A_2$ |
|---|---|---|---|
| **NMI** | 0.9326 | 0.9479 | 0.9482 |
| **$NMI_{max}$** | 0.8928 | **0.9457** | **0.9298** |
| **FNMI** | 0.7551 | **0.9304** | **0.8722** |
| **AMI** | 0.6710 | **0.7815** | **0.7533** |
| **VI** | -0.6996 | -0.5662 | -0.5503 |
| **FMeasure** | 0.8675 | 0.8782 | 0.8852 |
| **BCubed** | 0.8302 | 0.8431 | 0.8543 |
| **R** | 0.9827 | **0.9915** | **0.9901** |
| **AR** | 0.4911 | 0.5999 | 0.6213 |
| **J** | 0.3320 | 0.4329 | 0.4556 |
| **W** | **0.8323** | **0.6287** | 0.8010 |
| **D** | 0.4985 | 0.6042 | 0.6260 |
| **S&S** | 0.7926 | 0.8004 | 0.8262 |
| **CC** | 0.5376 | 0.6004 | 0.6371 |
| **CD** | -0.3193 | -0.2950 | -0.2802 |

seen. We see that NMI and $NMI_{max}$ almost always prefer the larger number of clusters. In contrast, Variation of Information and Rand usually prefer $k = 2$ (Rand prefers $k = 2$ in all cases).

### F.3 PRODUCTION EXPERIMENT

To show that the choice of similarity index may have an effect on the final quality of a production algorithm, we conducted an experiment within a major news aggregator system. The system aggregates all news articles to *events* and shows the list of most important events to users. For grouping, a clustering algorithm is used and the quality of this algorithm affects the user experience: merging different clusters may lead to not showing an important event, while too much splitting may cause the presence of duplicate events.

There is an algorithm $\mathcal{A}_{prod}$ currently used in production and two alternative algorithms $\mathcal{A}_1$ and $\mathcal{A}_2$. To decide which alternative is better for the system, we need to compare them. For that, it is possible to either perform an online experiment or make an offline comparison, which is much cheaper and allows us to compare more alternatives. For the offline comparison, we manually grouped 1K news articles about volleyball, collected during a period of three days, into events. Then, we compared the obtained reference partition with partitions $A_{prod}$, $A_1$, and $A_2$ obtained by $\mathcal{A}_{prod}$, $\mathcal{A}_1$, and $\mathcal{A}_2$, respectively (see Table 8). According to most of the indices, $A_2$ is closer to the reference partition than $A_1$, and $A_1$ is closer than $A_{prod}$. However, according to some indices, including the well-known $NMI_{max}$, NMI, and Rand, $A_1$ better corresponds to the reference partition than $A_2$. As a result, we see that in practical application *different similarity indices may differently rank the algorithms*.

To further see which algorithm better agrees with user preferences, we launched the following online experiment. During one week we compared $\mathcal{A}_{prod}$ and $\mathcal{A}_1$ and during another — $\mathcal{A}_{prod}$ and $\mathcal{A}_2$ (it is not technically possible to compare $\mathcal{A}_1$ and $\mathcal{A}_2$ simultaneously). In the first experiment, $\mathcal{A}_1$ gave $+0.75\%$ clicks on events shown to users; in the second, $\mathcal{A}_2$ gave $+2.7\%$, which clearly confirms that these algorithms have different effects on user experience and $\mathcal{A}_2$ is a better alternative than $\mathcal{A}_1$. Most similarity indices having nice properties, including CC, CD, and S&S, are in agreement with user preferences. In contrast, AMI ranks $\mathcal{A}_1$ higher than $\mathcal{A}_2$. This can be explained by the fact that AMI gives more weight to small clusters compared to pair-counting indices, which can be undesirable for this particular application, as we discuss in Section 5.

