# OpenReview forum: "Systematic Analysis of Cluster Similarity Indices: How to Validate Validation Measures"
_ICLR.cc/2021/Conference — Reject_

### Official Review · AnonReviewer1 · 2020-10-19
**Interesting comparative review**

**Rating:** 7
**Confidence:** 4

**Review:**

The authors propose a comparative review of cluster similarity indices through a theoretical approach prescribing in advance the required mathematical properties for a given application.
The problem is clearly stated, as well as motivation and impact, and reference list is rich and up-to-date. Overall I vote for acceptance, with a couple of remarks I list hereafter.

Pros:
- authors detect critical issues affecting many indices
- not-so-popular indices are suggested as interesting and valid alternatives to more widespread approaches
- analysis is mathematically sound and thorough

Cons:
- the main motivating question remain unsolved: although the landscape of measures is well described, the discussion should be clearer in helping the working researcher in choosing the right metric for his problem.
- the benchmark datasets, even the real world ones, are still somehow “artificial” and not really representative of the different situations that can be met nowadays; I would strongly recommend testing the studied algorithms on more complex data such as -omics datasets (e.g. from GEO, or single-cell sequencing), or weather radar data, or particle physics data, to greatly add value to the manuscript.
- there is no real novel material introduced here, and somehow no real learning involved: however, the analysis is clear and accurate, and, as a review, indeed interesting.

---

> ### Author Response · Authors · 2020-11-13
> **Response to Reviewer 1**
>
> Thank you very much for your positive feedback and suggestions!
>
> Regarding the comments:
> - Based on all suggestions, we will significantly extend the discussion section and give a more in-depth discussion on how to choose an index based on the properties of a given application. Additionally, following the suggestion of Reviewer 4, we will add a flow chart showing how such a decision can be made.
> - We are ready to perform an additional motivating experiment on a more realistic dataset. Do you have in mind any realistic publicly available clustering dataset where a reference (ground truth) partition is given? We are not very familiar with the types of datasets that are mentioned in the review, we tried to search for them, but could not find ones with reference partitions.
> - Regarding the novelty, we prefer to view the present work as a meta-analysis: we improve upon many existing works and unify them into a stronger result. To summarize the main new theoretical results, monotonicity and constant baseline hadn’t been properly formalized and rigorously analyzed before, while the asymptotic constant baseline with its subsequent analysis of biases for pair-counting indices is entirely novel. Though we do not propose any new indices, our results indicate that some indices that are rarely used (CC, CD, and S&S) dominate most popular indices, which we believe is an important insight.

---

> > ### Author Response · Authors · 2020-11-20
> > **We updated the paper**
> >
> > We’ve just updated the paper. In particular, we significantly extended Section 5 (Conclusion and discussion) to give a more in-depth discussion on how to choose an index based on the properties of a given application. We added a flow chart (Figure 2) illustrating how one can make a decision. We also added additional experiments demonstrating the significance of the constant baseline requirement (see Section D.4).
> >
> > If there are any more comments or questions, we will be happy to address them!

---

### Official Review · AnonReviewer4 · 2020-10-26
**This paper aims to answer a very important and difficult question ....**

**Rating:** 7
**Confidence:** 4

**Review:**

##########################################################################
Summary:
This paper aims to answer a very important and difficult question, i.e., given a clustering application what are the desirable qualities (i.e., similarity indices) to have. This work argues that there are so many clustering similarity indices with (sometimes) disagreements among them. The authors run experiments on 16 real-world datasets and 8 well-known clustering algorithms and provide a theoretical solution and a list of desirable properties that can help practitioners make informed decisions. Moreover, the authors also discuss the important pros and cons of the similarity indices in the context of the applications.


##########################################################################
Reasons for score:
My vote for this paper is that it is marginally above the acceptance threshold. The main reasons for decisions are i) this work tries to answer a very important question especially for practical applications because there are so many clustering algorithms and similarity indices, and oftentimes, it is not known what is best for the given application. ii) Although the paper does provide a theoretical solution and pros and cons of the algorithms and indices, yet it fails to answer the primary question, i.e., what is best for a given application.

##########################################################################Pros:

1. This work tries to answer a very important question especially for practical applications because there are so many clustering algorithms and similarity indices, and oftentimes, it is not known what is best for the given application.
2. It provides insights (i.e., pros and cons) into the similarity indices based on the experiments on 16 real-world datasets and 8 well-known clustering algorithms.
3. This paper can also serve as an excellent “survey paper” on the clustering algorithms and similarity indices.

##########################################################################
Cons:

1. This work fails to answer the primary question, i.e.; What are the best similarity indices for a given application. I understand that there is no clear answer to this question, which makes this work interesting. But, the authors should have concluded the answer. For example, authors could have provided with a flow chart, i.e., if A is more desirable than B in the application, choose C, and so on. It is just an idea, the authors can come up with some more interesting way to communicate their findings that potentially answers the key question.
2. I am not sure whether this kind of answer can be provided without a thorough discussion section, which is currently very short.
3. Other than the above, the length of the paper (27 pages) is a strength (i.e., a standalone paper and a great survey paper), and weakness at the same time (i.e., so much redundant information that is readable otherwise).

##########################################################################
Questions during the rebuttal period:

The authors can respond to all the concerns. But, my major concern is #1 in the cons section. Moreover, I feel it is a very strong survey paper, but I'm not sure whether ICLR accepts survey papers.

---

> ### Author Response · Authors · 2020-11-13
> **Response to Reviewer 4**
>
> Thank you very much for your positive feedback and helpful comments!
>
> Regarding the comments:
> - We agree that adding a flow chart would better answer the main question, thank you for this suggestion. We are currently working on such an illustration and will add it to the updated paper. We hope that this would also lead to a clear understanding of the differences between the indices and their usefulness in a given application.
> - We will extend the discussion section and give a more in-depth discussion about the situations where the proposed properties are important and in what aspects good indices differ from each other.
> - Regarding the comment about survey papers, we like to view the work more like a meta-analysis as we improve on many existing works and unify them into a stronger result. Importantly, all our theoretical results are novel and, to the best of our knowledge, cannot be found in the literature. Though we do not propose any new indices, our results indicate that some indices that are rarely used (CC, CD, and S&S) dominate most popular indices, which we believe is an important conclusion.

---

> > ### Author Response · Authors · 2020-11-20
> > **We updated the paper**
> >
> > We’ve just updated the paper. In particular, we added a flow chart (Figure 2) illustrating how one can make a decision given a given application. Also, we significantly extended Section 5 (Conclusion and discussion) to provide a more in-depth discussion on how to choose an index based on the properties of a given application.
> >
> > If there are any more comments or questions, we will be happy to address them!

---

### Official Review · AnonReviewer2 · 2020-10-28
**Interesting paper on the challenging cluster evaluation problem**

**Rating:** 6
**Confidence:** 3

**Review:**

*Summary*

This papers presents a systematic analysis of clustering validation measures. Authors present evidences that validation measures for clustering can disagree; then they present a set of desirable properties that a validation measure should satisfy; finally, for a quite large number of literature measures, they show which are the properties satisfied by each of them.



*Positive points*

-> Validation of clustering is still an open issue which deserves attention by researchers
-> Authors show with synthetic and real examples that different validation measures may prefer different solutions
-> The analysis of the properties is interesting
->The paper is well written and easy to follow



*Negative points/questions/suggestions*

-> While the topic of cluster validation is worth to be investigated, the study is focused only on external validation, namely on the validation of clustering techniques when the true clustering is known. Even if being interesting, this does not represent the true challenge of cluster validation in real scenarios, which, in my opinion, is represented by the internal validation: given a true clustering problem (i.e. without the ground truth), how can I choose a clustering algorithm?
Even if authors provide an example of the real utility of this analysis (major news aggregator system), I think that external validation remains mainly used to compare algorithms.  I would suggest authors to include more comments on the practical usability of this analysis, maybe listing some other scenarios in which it can have a practical impact.


-> The whole analysis is done by assuming that the number of clusters can vary: authors use techniques which i) select automatically the number of clusters or ii) receive in input different cluster numbers. What would be the conclusions if the number of clusters is kept fixed (and equal to the true number)? Would the disagreements be so large also in this case? Knowing the number of clusters is a classical assumption in many different practical scenarios.

-> Concerning the properties, I think they are all reasonable (especially the “constant baseline” one), except the symmetry. As recognized also by authors, typically these measures are used to compare the result of a clustering algorithm to the ground truth, so being naturally asymmetric. Please consider to remove this property from the list of desirable properties.

-> Finally, I think that the last section can be largely improved. Actually Section 5 contains a summary of the findings, i.e. a summary of the properties fulfilled by the different validation measures. Some practical suggestions on how to choose a given measure given the specific applications (with practical examples) would increase the value of the work.

---

> ### Author Response · Authors · 2020-11-13
> **Response to Reviewer 2**
>
> Thank you very much for the positive feedback and helpful comments!
>
> First, let us mention some practical examples of external validation:
> - Our example with a news aggregator is a real production system where external validation is used.
> - We believe that the above situation is quite common. Often, there is no ground truth partition available, but it is relatively cheap to let a group of experts annotate a subset of the documents. This, combined with a proper validation index, tells us which method performs best. Such a pipeline is standard in many practical applications.
> - In topic modeling, algorithms can be compared based on an expert annotation of topics (e.g., [1]).
> - Similarly, there is a matching of web-documents with expert-annotated classes (e.g., [2]).
> - There are many services that group photos based on personality (e.g., Microsoft Photos app). In such cases, algorithms can be compared on a subset annotated by assessors.
> - Reference partitions are also used in other applications, like classification of network traffic (e.g., [3]).
> - Also, when a reference (ground truth) partition is available, external validation measures can be used to validate internal measures.
>
> Additionally, let us note that all pair-counting external measures can be used for graph-partition comparison, as we shortly discuss in the first paragraph of Section B.3. So, our theoretical results can be applied in such cases too.
>
> However, we agree that studying internal measures is an important (but different) topic, and some work has already been done in this direction. For instance, there is an influential paper by Kleinberg [4] proving an impossibility theorem: there are three simple and natural constraints such that no internal clustering measure can satisfy all of them. Some works further studied the topic [5], while others analyzed internal quality measures for community detection in graphs [6].
>
> The comment about fixing the number of clusters raises an important point: many biased indices have a bias towards either fine- or coarse-grained clusterings. Our results indicate that the number of clusters is not the best measure of granularity of clusterings. For pair-counting indices, we prove that the appropriate measure of cluster granularity is the number of inter-cluster pairs. Hence, this bias will still be present even when fixing the number of clusters. Indices like Jaccard will favor clusterings with heterogeneous cluster-sizes (fewer inter-cluster pairs), while Rand will usually favor clusterings with equally sized clusters (more inter-cluster pairs). Note that this bias would disappear if we fix all cluster sizes, but we are not aware of any applications where this assumption can be made. We will improve the discussion on biases in the main text accordingly and will also perform an illustrative experiment: we will generate random partitions into two clusters but with different sizes and show how biased indices behave.
>
> Let us also stress that we do not suggest that each property is desirable for each application. For instance, similarity can be desirable in some applications and not desirable in others. Similarly, the distance property is often not needed. But it is important to be aware of different properties of an index to avoid incorrect conclusions. We will extend the discussion section and give more examples and intuition.
>
> Regarding practical suggestions on how to choose an index given a specific application - we are going to significantly extend the discussion section and, in particular, follow the suggestion of Reviewer 4 and add a flow chart showing how one can choose an index based on an application.
>
> [1] Virtanen S., Girolami M. “Precision-Recall Balanced Topic Modelling”, Advances in Neural Information Processing Systems, 2019.
>
> [2] Wibowo W., Williams H. E. “Strategies for minimising errors in hierarchical web categorisation”, Proceedings of the eleventh international conference on Information and knowledge management, 2002.
>
> [3] Erman J., Arlitt M., Mahanti A. “Traffic classification using clustering algorithms”,  Proceedings of the 2006 SIGCOMM workshop on Mining network data, 2006.
>
> [4] Kleinberg J. M. “An impossibility theorem for clustering”, Advances in neural information processing systems. 2003.
>
> [5] Ben-David S., Ackerman M. “Measures of clustering quality: A working set of axioms for clustering”, Advances in neural information processing systems, 2008.
>
> [6] Van Laarhoven T., Marchiori E. “Axioms for graph clustering quality functions”, The Journal of Machine Learning Research, 2014.

---

> > ### Author Response · Authors · 2020-11-20
> > **We updated the paper**
> >
> > We’ve just updated the paper.
> > - We added more motivation regarding external measures to the introduction.
> > - The notion that fixing the number of clusters does not solve the constant baseline problem is illustrated in Figure 4 (Section D.4).
> > - We also significantly extended Section 5 (Conclusion and discussion) - we discuss how one can choose a suitable index and give some practical examples. Figure 2 illustrates how one can make a decision.
> >
> > If there are any more comments or questions, we will be happy to address them!

---

### Official Review · AnonReviewer3 · 2020-10-29
**Nice paper that is of interest to the community. However, the novelty of the paper is not clear.**

**Rating:** 7
**Confidence:** 3

**Review:**

Cluster Similarity Indices (CSIs) take as input two clusterings A, B and assign a similarity score for the given pair of clusterings. The index calculates a score based on the number of pairs of elements that clustered together on both clustering (N++), those that are not clustered together in non of A,B (N--), those that are clustered together in A but not in B (N+-), and vice versa (N-+). CSIs can be used to evaluate clusterings produced by different algorithms with respect to some reliable reference clustering on a single instance, and choose the one that is the closest to the reference clustering (indicated by the CSIs). The selected clustering algorithm can be then applied to different instances of the same kind where we do not have a reference clustering.

The authors motivate their problems in two ways. First, they consider a set of clusterings on different instances (that also contain a reference clustering) and show that the different indices disagree in the ranking of the clusterings, showing that it is not straight-forward to choose a good clustering index for an application. Then, they conduct a live experiment where they want to replace an existing clustering with one of two new available clusterings in an application, and show that the clustering that was preferred by most indices was the one that indeed showed better performance for the specific application.

The focus of this paper is to provide a way of comparing different similarity indices. In particular they provide a set of desirable properties to characterise a CSI and show which of the considered CSIs respect have those properties. The properties that the authors consider follow an intuitive sense of a good clustering. Among the properties that the authors consider they emphasize on what they call Constant Baseline. Roughly speaking, Constant Baseline requires that if one compares a clustering A to any random partitioning of the elements into clusters B, then the index should assign a minimal similarity score between A and B. Notice that the random partitioning might have any distribution of cluster sizes.
Several indices are considered and analysed with respect to the defined properties, and the authors advocate the use of two specific indices that satisfy all but one of the properties that they define (namely, the metric property).

=====

PROS
+ Well motivated problem and relevant to the ML community
+ The authors conduct a live experiment verifying the importance of CSIs in real-world applications.
+ The authors formulate a property that deals with a previously observed bias of CSIs toward clusterings with high-cardinality clusters.

CONS
- This is potentially an incremental work. The characterization of CSIs with respect to some set of desired properties is not new, see e.g., [1] (some of the properties are shared).

Given the plethora of the clustering algorithms that exist out there it is important to have a systematic way of choosing which clustering algorithm one should choose. In general, I think that the problem is well-motivated.

This is not the first time that properties of CSIs are considered for the selection of a clustering algorithm. E.g., [1] did a similar study. From the discussion in Appendix A, the authors mention "In the current research, we give a more comprehensive list of constraints and focus on those that are desirable in a wide range of applications". Are there some applications where the axioms from [1] are not applicable? What is the added value compared to [1], and compared to other similar studies? Do they have properties/axioms that deal with the reported bias of many indices to clusterings with higher cardinality clusters? I would like to understand these differences to assess more properly the novelty of this paper.

One experiment that would make the claim for the significance of the Constant Baseline stronger would be to compare the reference clusterings from the second experiments in Section 3 to two randomly generated partitions, one with higher cardinality clustering than the other, and show that indeed, the CSIs that satisfy the Constant Baseline property have a more-or-less equal score to the two clustering, while the CSIs that do not satisfy it tend to prefer the clustering with larger clusters. Such an experiment would suggest that this is not just a theoretic property that is nice to have, but it also makes a significant difference in practice.

Overall, the paper is nicely written and easy to follow. I tend to be positive. My main concern is on the comparison with related work and the novelty of the paper, that I don't fully understand.

[1] Enrique Amigo, Julio Gonzalo, Javier Artiles, and Felisa Verdejo. A comparison of extrinsic ´clustering evaluation metrics based on formal constraints. Information retrieval, 12(4):461–486,2009.

---

> ### Author Response · Authors · 2020-11-13
> **Response to Reviewer 3**
>
>
> Thank you very much for the positive feedback and helpful comments!
>
> Regarding previous works on comparative analysis of cluster similarity indices, we see that the problem is yet not completely solved and deserves more attention and a more thorough analysis. In practical applications, clusterings are still compared using NMI, Jaccard, and many other indices that can be biased in different ways. We believe that our work significantly improves over previous attempts and gives a more thorough and rigorous analysis.
>
> In particular, let us discuss the differences with [1] in more detail. In [1], four ‘constraints’ are proposed: ‘cluster homogeneity’ is a weaker analog of our ‘monotonicity w.r.t. perfect splits’, ‘cluster completeness’ is equivalent to our ‘monotonicity w.r.t. perfect merges’, so we give a deeper analysis of this monotonicity property. The third ‘rag bag’ constraint is motivated by the subjective claim that ‘introducing disorder into a disordered cluster is less harmful than introducing disorder into a clean cluster’. This is argued to be important for their particular application (text clustering), but we found no other work that deemed this constraint necessary; hence we disregarded this one.
> The last constraint is interesting since it concerns the balance between making errors in large and small clusters. However, it poses a particular balance while we think that this balance may differ per application, and we are not aware of a proper formalization of “the level of balance”. We will add a discussion about that to the conclusion section.
> More importantly, biases and constant baseline are not discussed in [1] while they do mention it as a promising direction in their conclusion. The constant baseline is a particular focus of our study, we find it extremely important, and this requirement rejects most of the widely used indices.
> Finally, [1] considers a smaller number of indices and properties, and their analysis is less formal (some claims are not proved, but demonstrated via experiments).
> We will extend our related work section to include a more in-depth comparison with [1].
>
> Additionally, we will extend the related work section with more discussions on constant baseline. While this property is not covered in [1], its importance was discussed in several papers (Albatineh et al., 2006; Lei et al., 2017; Romano et al., 2014; Strehl, 2002; Vinh et al., 2009; 2010). However, to the best of our knowledge, this property was not formalized and was not rigorously analyzed for all indices widely used in the literature. Having this analysis is important for understanding the indices.
>
> Regarding the experiment illustrating that violating constant baseline may lead to significant bias towards some cluster sizes, we agree that it can be useful and will make such an experiment. We are planning to generate random partitions with different numbers of clusters and show the dependence of CSIs values on the number of clusters. As a reference partition, we are planning to take the true clusters from the last experiment in Section 3. Is this the one you suggested?

---

> > ### Author Response · Authors · 2020-11-20
> > **We updated the paper**
> >
> > We’ve just updated the paper.
> > - We significantly extended the related work section (Appendix A) to highlight the differences with Amig´o et al. and our novelty regarding the constant baseline
> > - The comment regarding additional experiments for the constant baseline property is addressed in Appendix D.4, Figure 3. Here we show the dependence of all indices on the number of clusters and their balance.
> > - We also significantly extended Section 5 (Discussion and conclusion).
> >
> > If there are any more comments or questions, we will be happy to address them!

---

### Author Response · Authors · 2020-11-24
**Summing up our updates**

According to the reviewers’ comments, we updated the paper.
- We significantly extended Section 5 (conclusion and discussion) – to answer the question “how one can choose a suitable index given a particular application” based on theoretical results obtained in our paper.
- We conducted additional experiments (Appendix D.4, Figures 3 and 4) to answer the questions about the practical importance of constant baseline and what happens if we fix the number of clusters.
- We also extended the related work section (Appendix A) to clarify the novelty of our research compared to previous studies.

We hope that we address all the concerns and will be glad to get feedback from the reviewers. If there are any more comments or questions, we will be happy to address them!

---

### Decision · Program_Chairs · 2021-01-07
**Final Decision**

**Decision:**

Reject

**Comment:**

The paper goes over a long list of proposed clustering similarity indices and attempts
to provide a taxonomy of those by their different approaches and the extent by which they
satisfy a list of "desired properties" proposed by the authors.
This is very much in the spirit of earleir work on clustering similaritie by [Meila 2007]
and on clustering quality measures [Ackerman, Ben-David 2008, 2009].

While there may be some interest in such a compendium, there is no much novelty in this paper
and it relevance to practice is also unclear.

---

> ### Author Response · Authors · 2021-01-14
> **All concerns are addressed in the paper**
>
> Needless to say, we are disappointed and surprised with this decision, taking into account positive feedback from all the reviewers and a productive discussion period, which led to an improved paper and increased scores. Unfortunately, we cannot contest the decision, but we would like to reply to the concerns:
> - **Novelty.** Our paper is theoretical. In particular, we are the first to develop a framework for rigorous analysis of constant baseline for general and pair-counting indices (Section 4.6). Furthermore, our methodology advises using different indices than that are currently being used in practice. Also, note that each cell in Tables 2 and 3 corresponds to rigorous proof. Some of them are simple, but many are quite complicated, and most of them are new. Hence, we cannot agree that the paper does not have much novelty.
> - **Relevance to practice.** Section 3 addresses that. We clearly demonstrate that choosing a “wrong” similarity index could lead to completely opposite conclusions (see, e.g., Table 1 that shows huge disagreement between popular indices). Our work aims to attract more attention to this problem and encourage researchers and practitioners to use “better” indices, which is essential for the whole field of clustering. We also conducted a practical experiment with real multiuser services showing that a flawed metric could mislead optimization of the system in the wrong direction. Notably, the best index (according to our framework) has previously been overlooked by practitioners.
> - **Importance.** Let us stress that cluster similarity measures are often chaotically used, and our paper aims to bring order to this area, which we believe is important.
> - **[Meila 2007]** advocates using the Variation of Information. While also using a theoretical approach, that paper does not consider the constant baseline property, which is extremely important and is an essential part of our contribution. As a result, [Meila 2007] comes to a completely different conclusion compared to our work. (We additionally discuss theoretical differences with this paper in Section A).
> - **[Ben-David and Ackerman, 2009]** is indeed similar in spirit. But it analyzes a different problem (internal cluster evaluation) and has completely different results, approaches, etc. We discuss internal cluster evaluation and refer to this paper in Sections 1 and A.
>
> To sum up, we do not understand the reason for rejection. We would appreciate any additional details on your concerns about novelty and practical importance, considering our comments and theoretical contribution.